# Repressed *Blautia*-acetate immunological axis underlies breast cancer progression promoted by chronic stress

Ling Ye [1,6], Yuanlong Hou[2,3,6], Wanyu Hu[1,6], Hongmei Wang [4], Ruopeng Yang[1], Qihan Zhang [5], Qiaoli Feng [5], Xiao Zheng [2], Guangyu Yao [5] ✉ & Haiping Hao [2] ✉

Chronic stress is a known risk factor for breast cancer, yet the underlying mechanisms are unclear. This study explores the potential involvement of microbial and metabolic signals in chronic stress-promoted breast cancer progression, revealing that reduced abundances of *Blautia* and its metabolite acetate may contribute to this process. Treatment with *Blautia* and acetate increases antitumor responses of CD8+ T cells and reverses stress-promoted breast cancer progression in female mice. Patients with depression exhibit lower abundances of *Blautia* and acetate, and breast cancer female patients with depression display lower abundances of acetate, decreased numbers of tumor-infiltrating CD8+ T cells, and an increased risk of metastasis. These results suggest that *Blautia*-derived acetate plays a crucial role in modulating the immune response to breast cancer, and its reduction may contribute to chronic stress-promoted cancer progression. Our findings advance the understanding of microbial and metabolic signals implicated in cancer in patients with depression and may provide therapeutic options for female patients with breast cancer and depression.

Breast cancer is a major global health issue, comprising about 25% of all cancer cases in women[1]. In 2020, approximately 2.3 million new cases were diagnosed worldwide, and it remains a leading cause of cancer-related deaths among women, accounting for an estimated 685,000 deaths in the same year[2]. Depression, a mental health condition, has been associated with the development, progression, and poor treatment outcomes of breast cancer based on clinical and preclinical studies[3]. Comorbid depression can promote breast cancer progression[4–7], and breast cancer patients are susceptible to developing depression due to chronic psychological stress, leading to poor prognosis and high mortality[8].

Depression's impact on the immune system and inflammation pathways may contribute to breast cancer development. Previous studies have focused on how the activation of the hypothalamic–pituitary–adrenal axis and the sympathetic nervous system from depression leads to the repression of tumor immunology, which may contribute to facilitated growth and metastasis of cancer[9–12]. It has been suggested that β-adrenergic blockers may help restore chronic stress-repressed tumor immunology[13], but there is no direct evidence yet of the combinatorial use of β-adrenergic blockers with anti-cancer drugs improving the survival of cancer patients with comorbid depression[14,15]. Therefore, there is an urgent need to uncover

[1]NMPA Key Laboratory for Research and Evaluation of Drug Metabolism, Guangdong Provincial Key Laboratory of New Drug Screening, School of Pharmaceutical Sciences, Southern Medical University, Guangzhou 510515, China. [2]State Key Laboratory of Natural Medicines, Jiangsu Province Key Laboratory of Drug Metabolism, China Pharmaceutical University, Nanjing 210009, China. [3]Department of Pharmacy, Shenzhen Luohu People's Hospital, Shenzhen 518000, China. [4]Department of Radiation Oncology, Nanfang Hospital, Southern Medical University, Guangzhou 510515, China. [5]Breast Center, Department of General Surgery, Nanfang Hospital, Southern Medical University, Guangzhou 510515, China. [6]These authors contributed equally: Ling Ye, Yuanlong Hou, Wanyu Hu. ✉e-mail: yaogy@smu.edu.cn; haipinghao@cpu.edu.cn

innovative causal links and mechanisms connecting depression to cancer progression and develop effective therapeutic approaches.

Gut microbiota and its metabolites have critical roles in shaping host immunity and are involved in diverse pathophysiological events of many diseases, including cancer and depression[16–18]. Studies have shown that chronic stress can alter the composition and metabolic features of gut microbiota, disrupting the immune system[18–21]. Therefore, dysbiosis of gut microbiota caused by chronic stress may play a critical role in cancer progression. However, it is still unclear how microbial and particularly the relative host–microbiota co-metabolic signals to the immune system are involved during chronic stress-promoted cancer progression. Notably, oral supplementation with *Akkermansia muciniphila* restored the efficiency of PD-1 blockade by expanding T cells[22], gut microbiota-mediated bile acid metabolism inhibited liver cancer progression by increasing NKT cell activity[23], and microbial short-chain fatty acids (SCFAs) enhanced the anti-tumor activity of CD8$^+$ T cells and improved cancer immunotherapy[24,25]. *Lactobacillus rhamnosus* JB-1, which possesses antidepressant-like effects in mice, has been found to modulate the immune system through the induction of immunosuppressive T regulatory cells[26,27]. Therefore, it is reasonable to hypothesize that establishing innovative causal links and mechanisms connecting gut microbiota to cancer progression could be beneficial in the development of effective therapeutic approaches for cancer patients with comorbid depression.

In this work, our study aims to elucidate the role of gut microbiota and its associated metabolic signals in the promotion of breast cancer progression within the context of depression. We develop a temporospatial microbe-phenotype triangulation approach, integrating metabolomics and metagenomics, to precisely identify the causal microbes and metabolites involved in the co-occurrence of depression and breast cancer. Our findings unveil that chronic stress leads to a reduction in the abundance of the key microbe *Blautia* and its metabolite acetate. These deficiencies may contribute to the promotion of breast cancer progression in the context of chronic stress by reducing the number and function of CD8$^+$ T cells in the tumor microenvironment. Additionally, we demonstrate the potential of *Blautia* and acetate treatment in enhancing the antitumor responses of CD8$^+$ T cells, thereby mitigating the effects of chronic stress on breast cancer progression.

## Results

### Chronic stress accelerates breast cancer progression associated with compromised CD8$^+$ T cell response

To investigate the impact of chronic stress on tumor progression, we exposed mice to chronic restraint stress (CRS) or chronic unpredictable stress (CUS) for 2 weeks (Supplementary Fig. 1a, b) before inoculating them with 4T1 or EO771 cells, resulting in the establishment of two distinct breast cancer growth models (Fig. 1a, b). Consistent with previous findings[28,29], we observed accelerated tumor progression in mice exposed to chronic stress when compared to non-stressed mice (Fig. 1c, d and Supplementary Fig. 1c, d). We also observed a significant decrease in T cells, particularly CD8$^+$ T cells, in the blood, spleen, and tumor tissue in the chronic stress conditions; however, no significant changes were observed in B cells, macrophages, monocytes, and NK cells in the tumor tissue (Supplementary Fig. 1e–g). Furthermore, we observed a decline in the ability of tumor-infiltrating CD8$^+$ T cells to produce inflammatory cytokines interferon-gamma (IFN-γ) in stressed mice, in addition to the decrease in their numbers (Fig. 1e–h), despite no significant changes were observed in the levels of cytolytic molecule granzyme B (GZMB) and tumor necrosis factor α (TNF-α) (Supplementary Fig. 1h). Our results suggest that chronic stress leads to accelerated breast cancer progression through compromised responses of infiltrated CD8$^+$ T cells in the tumor.

### Gut microbiota mediates chronic stress-promoted breast cancer progression

To explore the potential link between gut microbiota and breast cancer progression exacerbated by chronic stress, a cocktail of antibiotics (ABX) was administered to mice in order to deplete commensal bacteria in the gut (Supplementary Fig. 2a, b). The study showed that chronic stress accelerated breast cancer progression, whereas the tumor-promoting effects of chronic stress were absent in mice treated with ABX (Fig. 2a–c and Supplementary Fig. 2c–e). Notably, no significant differences were observed in the abundances and function of tumor-infiltrated CD8$^+$ T cells between stressed and non-stressed mice following ABX treatment (Fig. 2d). Moreover, the ABX treatment per se did not affect breast cancer progression in non-stressed mice compared to the group not receiving ABX (Fig. 2b–d). These findings suggest a potential role of gut commensal bacteria in mediating the tumor-promoting effects of chronic stress in breast cancer progression.

To further confirm the essential role of gut microbiota in chronic stress-promoted breast cancer progression, we established a co-housing mouse model (Fig. 2e), enabling the horizontal transfer of microbial communities from one mouse to another and, thereby the expected normalization of gut-associated phenotype. Our results demonstrated that the accelerated progression of breast cancer induced by chronic stress was abolished when the stressed tumor mice were co-housed with non-stressed mice (Fig. 2f, g and Supplementary Fig. 2f, g). More importantly, the abundance and function of tumor-infiltrating CD8$^+$ T cells were significantly enhanced in the stressed mice under co-housing conditions (Fig. 2h). Furthermore, increased production of IFN-γ by tumor-infiltrating CD8$^+$ T cells was also observed in co-housed stressed mice (Fig. 2h), although no significant differences were observed in the production of GZMB and TNF-α by tumor-infiltrating CD8$^+$ T cells (Supplementary Fig. 2h). Taken together, our data support the involvement of gut microbiota in the promotion of breast cancer progression under conditions of chronic stress.

### Declined *Blautia* underlies chronic stress-promoted breast cancer progression

To reduce the impact of confounding microbes and identify exact causal microbiota and metabolites, we employed and modified a temporospatial microbe–phenotype triangulation approach[30]. We initially screened for microbes that exhibited significant changes upon co-housing along the spatial axis, which involved analyzing cecum samples from different groups. We identified several microbes that were significantly altered, including *Blautia*, *Ruminiclostridium*, *Intestinimonas*, and *Peptococcus* (Fig. 3a–c and Supplementary Fig. 3e–h). To further verify the possible causal microbes, we investigated the abundance of microbes over time, and we reasoned that only the microbiota exhibiting stable time-dependent changes are truly causal. Therefore, we analyzed fecal samples from the co-housed stressed tumor group at different time points along the temporal axis. We observed a gradual increase in the abundance of *Blautia*, *Ruminiclostridium*, *Ruminococcus*, and *Lachnospiraceae_NK4A136_group* from day 14, with maximal changes seen after 40 days (Fig. 3d–f). This temporospatial approach allowed us to identify *Blautia* and *Ruminiclostridium* as potential causal microbes responsible for facilitating depression-related breast cancer progression.

To validate the initial findings, we conducted 16 S ribosomal RNA (rRNA) gene sequencing on cecal contents collected from non-stressed and stressed mice with or without breast cancer at the end of the experiments. Beta-diversity analysis utilizing the Bray–Curtis dissimilarity metric in principal coordinate analysis revealed distinct clustering of gut microbial profiles between the non-stressed and stressed groups (Supplementary Fig. 3a). Specifically, the stressed group exhibited a significant reduction in the abundance of *Blautia*

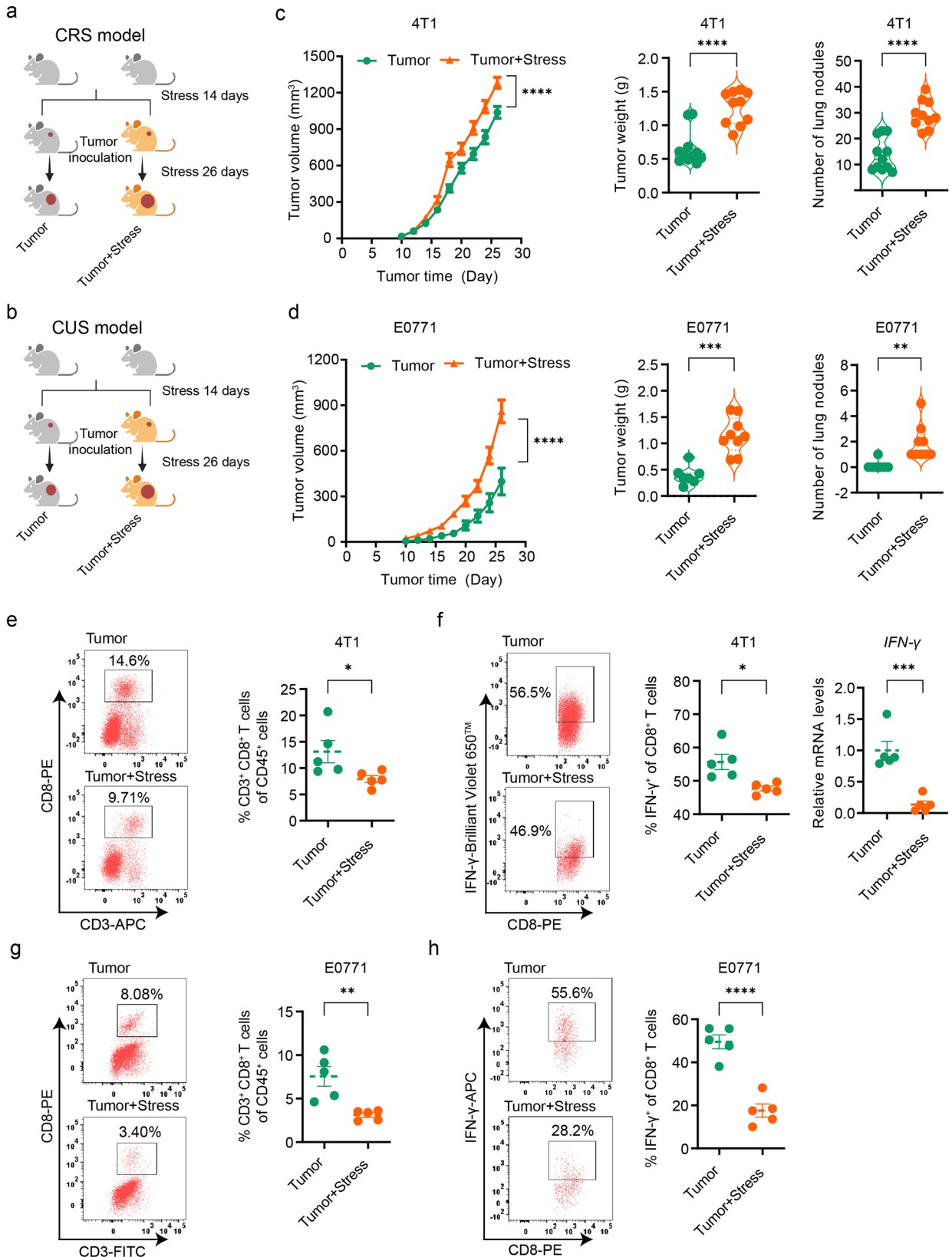

but not *Ruminiclostridium* compared to the non-stressed group (Supplementary Fig. 3b, c). Furthermore, we confirmed the association of *Blautia* spp. with chronic stress by assessing their abundance in the CUS model, where the levels of *Blautia coccoides* (BC) and *Blautia obeum* (BO) were significantly decreased (Supplementary Fig. 3d). Moreover, 16 S rRNA sequencing was performed on stool

samples obtained from patients with major depressive disorder (MDD) and healthy controls (HC), revealing a significantly lower abundance of *Blautia* in MDD patients compared to that in HC (Fig. 3g).

To confirm the causal role of *Blautia* in chronic stress-promoted breast cancer progression, we conducted an experiment in which BC

**Fig. 1 | Chronic stress promotes breast cancer progression and impairs intra-tumoral CD8⁺ T cells. a** Experimental design for Balb/c mice: one group was subjected to CRS for 14 consecutive days before inoculation with 4T1 cells, and the other group was inoculated with 4T1 cells without stress exposure. Created with BioRender.com. **b** Experimental design for C57BL/6 mice: one group was exposed to CUS for 14 consecutive days before inoculation with E0771 cells, and the other group was inoculated with E0771 cells without stress exposure. Created with BioRender.com. **c** Tumor growth volume, weight, and lung metastasis burden in mice with 4T1 cells inoculation ($n = 11$ for tumor group, $n = 10$ for tumor + stress group). **d** Tumor growth volume, weight, and lung metastasis burden in mice with E0771 cells inoculation ($n = 7$ for tumor group, $n = 9$ for tumor + stress group). **e** Tumor-infiltrating CD8⁺ T cells in mice with 4T1 cells inoculation were analyzed by flow cytometry ($n = 5$ per group). **f** IFN-γ expression in tumor-infiltrating CD8⁺ T cells of mice with 4T1 cells inoculation analyzed by flow cytometry or qPCR ($n = 5$ per group). **g** Tumor-infiltrating CD8⁺ T cells in mice with E0771 cells inoculation were analyzed by flow cytometry ($n = 5$ per group). **h** IFN-γ expression in tumor-infiltrating CD8⁺ T cells of mice with E0771 cells inoculation analyzed by flow cytometry ($n = 5$ per group). Data were presented as mean ± SEM. Statistical significance was determined using two-way ANOVA followed by Sidak's multiple comparison test for tumor volume and unpaired two-tailed Student's $t$-test for the other data. Significance levels are denoted as $*p < 0.05$; $**p < 0.01$; $***p < 0.001$; $****p < 0.0001$. "ns" indicates no significant difference. Source data and exact $p$ values are provided in the Source data file.

or BO was administered to tumor mice subjected to chronic stress (Fig. 4a, b). The results demonstrated that treatment with both BC and BO significantly inhibited the accelerated progression of breast cancer induced by chronic stress (Fig. 4c, d). Moreover, compared to the stressed control group, treatment with both BC and BO substantially increased the infiltration of CD8⁺ T cells into the tumor and effectively stimulated the production of IFN-γ by CD8⁺ T cells (Fig. 4e, f).

These findings provide strong evidence supporting the essential role of *Blautia* in the link between chronic stress and the progression of breast cancer and suggest that targeting *Blautia* could potentially serve as a therapeutic strategy for breast cancer patients experiencing chronic stress.

### Reduced *Blautia* leads to compromised production of acetate

To elucidate the mechanism by which the reduced abundance of *Blautia* is involved in the progression of breast cancer promoted by chronic stress, we conducted a metabolomic analysis using serum and tumor samples obtained from mice involved in co-housing experiments. Since microbial metabolites are recognized as the dominant signals in conveying host-microbiota interactions[16,31], we next extended to identify the relevant microbial metabolites associated with the causal microbial changes. Principal component analysis (PCA) revealed a distinct clustering of metabolites in different groups (Fig. 5a, b), and a total of 203 and 208 metabolites were identified in serum and tumor tissue, respectively. Notably, the heatmap showed 50 discriminant metabolites among the three groups, indicating that co-housing exerted a profound effect on shaping microbe-host metabolic balance (Fig. 5c, d). Acetate and stearoylcarnitine were identified as the most significantly changed metabolites in serum and tumor samples after co-housing (Fig. 5e, f and Supplementary Fig. 4c, d). Subsequently, we conducted a correlation analysis between the metabolites and gut microbiota and found that the contents of acetate and stearoylcarnitine were strongly and positively correlated with the abundance of *Blautia* (Supplementary Fig. 4a, b). Previous findings have shown that members of *Blautia* are prolific producers of acetate[32]. Our results revealed that the supplemented strains, BC and BO, were also capable of producing large amounts of acetate (Supplementary Fig. 4e), thereby supporting their effects in abrogating chronic stress accelerated cancer progression. Consistently, the content of acetate in tumor tissues was found to be reduced in the CUS model (Fig. 5g). Furthermore, the content of acetate in MDD patients was also significantly lower compared to that in the HC (Fig. 5h).

To further verify the effect of the *Blautia*-acetate axis, we used shotgun metagenomics to determine the alteration of the enzymes of *Blautia* involved in acetate metabolism. Our metagenomic data indicated that the Wood–Ljungdahl pathway, which is the main pathway generating acetate by *Blautia*[32], was significantly downregulated, suggesting decreased acetyl-CoA capacity (Fig. 5i, j). We observed that nearly all the key genes involved in $CO_2$ fixation (K00198, K15022), formyl-tetrahydrofolate ligase (K01491, K00297), and acetyl-CoA synthase (K01438) were significantly reduced (Fig. 5k),

supporting the reduction of acetyl-CoA biosynthesis upon chronic stress. Further analysis revealed striking changes in pyruvate metabolism including phosphoacylase (K00625) and acetate kinase (K00925) in stressed tumor mice (Fig. 5k). To identify the bacterial genera involved in Wood–Ljungdahl pathway, we conducted an analysis of species contributing to K00198 from the top 50 bacterial genera (Supplementary Fig. 4f), and it was determined that the crucial K00198 was present in *Blautia*, *Lachnoclostridium*, *Ruminococcus*, *Clostridium*, and *Oscillibacter genera*. Taken together, these findings support that, under conditions of chronic stress, the decreased abundance of *Blautia* is more likely to contribute to the reduction of acetate production and that the disruption of the *Blautia*-acetate axis is likely involved in the pathological progression of breast cancer promoted by chronic stress.

### Acetate activates CD8⁺ T cells and retards chronic stress-promoted breast cancer progression

To verify the causal relationship between acetate and the progression of breast cancer under chronic stress, we conducted experiments involving the supplementation of sodium acetate (Fig. 6a). Our results demonstrate that sodium acetate treatment significantly suppressed tumor progression in stressed tumor mice when compared to sodium chloride treatment (Fig. 6b, c and Supplementary Fig. 5a, b). Notably, we observed a marked increase in intratumoral accumulation of CD8⁺ T cells and IFN-γ production in the stressed tumor mice following supplemental sodium acetate (Fig. 6d, e). Interestingly, our findings suggest that sodium acetate treatment did not yield a significant inhibition of breast cancer progression in non-stressed tumor mice (Fig. 6b, c).

To investigate the essential role of CD8⁺ T cells in the tumor retardation mediated by sodium acetate in stressed tumor mice, we employed a CD8 neutralizing antibody to deplete CD8⁺ T cells in mice. Our results demonstrated that depletion of CD8⁺ T cells abolished the antitumor effect of sodium acetate (Fig. 6f–h), indicating the crucial role of CD8⁺ T cells in tumor retardation by sodium acetate in stressed tumor mice. To further examine the effect of acetate on CD8⁺ T cells, we evaluated the survival and functionality of CD8⁺ T cells treated with sodium acetate in vitro. Consistent with previous studies[25], our data demonstrated that, unlike sodium propionate and sodium butyrate, sodium acetate did not directly impact the survival and functionality of CD8⁺ T cells (Supplementary Fig. 5c, d). We also assessed the direct effect of sodium acetate on the survival of breast cancer cells and found no evidence that sodium acetate can induce breast cancer cell death (Supplementary Fig. 5e). Therefore, we established a cell model where CD8⁺ T cells were co-cultured with 4T1 cells to investigate the effect of sodium acetate. Interestingly, the cytotoxic capacity of CD8⁺ T cells co-cultured with 4T1 cells was significantly enhanced by sodium acetate treatment in a dose-dependent manner (Fig. 7a). Notably, sodium acetate treatment, along with sodium propionate and sodium butyrate, increased the survival and production of IFN-γ in CD8⁺ T cells co-cultured with 4T1 cells (Fig. 7b, c and Supplementary Fig. 5f–h). Our findings suggest that

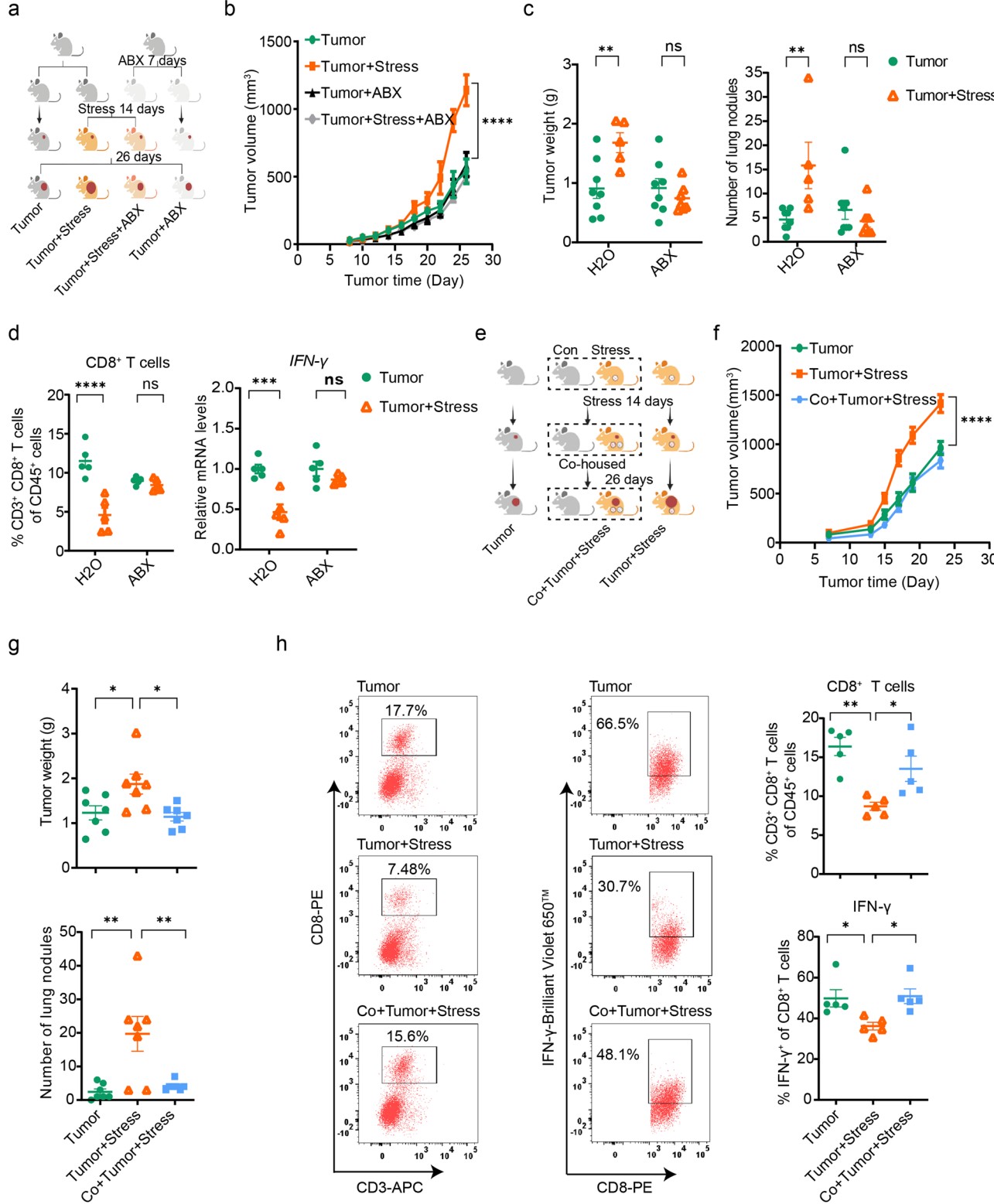

acetate may not have direct effects on cancer cells and CD8+ T cells, but it can prevent impairment of CD8+ T cells induced by cancer cells, thereby boosting anti-tumor immunity.

To explore how acetate impacts CD8+ T cell function in co-culture conditions, we performed RNA sequencing on CD8+ T cells co-cultured with 4T1 cells treated with or without sodium acetate. Of the 29,956 gene transcripts expressed, only 1828 genes (6.1%) differed significantly between the groups, and no significant cluster was observed between groups (Fig. 7d). Of the 28 effector-, memory-, and

exhaustion-related genes of CD8+ T cells[25], the mRNA levels of *Gzma*, *Lta*, *Tnf*, *Batf3*, *Eomes*, *Foxo1*, *Id2*, *Irf4*, *Tbx21*, *Il12rb2*, *Cd69*, *Elf4*, *Nfatc3*, *Tcf3*, *Il7r*, and *Il27ra* showed significant difference, but the log2 fold-change in gene expression >1 was only observed for *Gzma* and *Lta* (Supplementary Fig. 5i). Neither glycolysis nor tricarboxylic acid (TCA) cycle genes clustered differently between acetate-treated and control CD8+ T cells. Furthermore, except for the decreased expression of some genes in the glycolytic pathway (*Gpi1*, *Hk1*, *Pgam1*, *Pgk1*, *Pkm*, *Tpi1*, and *Pfkp*) (Supplementary Fig. 5j), sodium acetate treatment had

**Fig. 2 | Gut microbiota modulates breast cancer progression exacerbated by chronic stress. a** Experimental design for the ABX treatment: mice were treated with sodium chloride or ABX for 7 days prior to dividing into two groups. One group was exposed to chronic restraint stress for 14 consecutive days before 4T1 cells inoculation and continued until the end of the experiment, while the other group was inoculated with 4T1 cells without stress exposure. The ABX groups received ABX throughout the experiment. Created with BioRender.com. **b** Tumor growth volume in the ABX treatment experiment (tumor group: $n = 8$, tumor + stress group: $n = 5$, tumor + ABX group: $n = 8$, tumor + stress + ABX group: $n = 7$). **c** Tumor weight and tumor burdens on the lung in the ABX treatment experiment (tumor group: $n = 8$, tumor + stress group: $n = 5$, tumor + ABX group: $n = 8$, tumor + stress + ABX group: $n = 7$). **d** Tumor-infiltrating CD8[+] T cells and abundance of IFN-γ in tumor-infiltrating CD8[+] T cells in the ABX treatment experiment ($n = 5$

per group). **e** Experimental design for the co-housing experiment, in which mice with non-stressed, stressed, and co-housed stressed conditions were inoculated with 4T1 cells, respectively. Created with BioRender.com. **f** Tumor growth volume in the co-housing experiment ($n = 7$ per group). **g** Tumor weight and tumor burdens on the lung in the co-housing experiment ($n = 7$ per group). **h** Tumor-infiltrating CD8[+] T cells and IFN-γ abundance of tumor-infiltrating CD8[+] T cells in the co-housing experiment ($n = 5$ per group). Data were presented as mean ± SEM. Statistical significance was determined using two-way ANOVA followed by Sidak's multiple comparison test for (**b–f**) and one-way ANOVA followed by Tukey's multiple comparisons test for (**g, h**). Significance levels are denoted as *$p < 0.05$; **$p < 0.01$; ***$p < 0.001$; ****$p < 0.0001$; "ns" indicates no significant difference. Source data and exact $p$ values are provided in the Source data file.

---

no significant influence in TCA cycle genes at the individual gene level (Supplementary Fig. 5k). Several monocarboxylate transporters including SLC16A1 and SLC16A3 were involved in acetate transport[33,34]. At the mRNA level, we observed that the expression of *Slc16a1* but not *Slc16a3*, was increased in acetate-exposed CD8[+] T cells (Fig. 7e, f). Previous studies have demonstrated that acetate may enhance IFN-γ production in a manner dependent on acetyl-CoA synthetase (ACSS) and ATP citrate lyase (ACLY)[33,34]. In our study, we observed the upregulation of mRNA levels of *Acss1, Acss2,* and *Acly* in acetate-exposed CD8[+] T cells (Fig. 7f). Isotopic tracing studies revealed that acetate contributed to the production of citrate, oxoglutarate, fumarate, and malate (Fig. 7g). Moreover, exogenous acetate increased the pool of cellular acetyl-CoA available to CD8[+] T cells (Fig. 7h). Together, these data suggest that in conditions of CD8[+] T cells co-culture with cancer cells, sodium acetate treatment may boost the activity of CD8[+] T cells via restoring the acetyl-CoA pool and subsequently augment IFN-γ production.

We further conducted a study to investigate the relationship between serum acetate concentration, tumor-infiltrating CD8[+] T cells, and depression in breast cancer patients. Our results showed that breast cancer patients with depression had lower levels of serum acetate compared to those without depression (Fig. 8a), as well as lower levels of tumor-infiltrating CD8[+] T cells (Fig. 8b). Additionally, there was a positive correlation between acetate levels and tumor-infiltrating CD8[+] T cells (Fig. 8c), while depression showed a negative correlation with acetate levels (Fig. 8d). Moreover, breast cancer patients with depression had a significantly lower metastasis-free survival rate compared to breast cancer patients without depression (Fig. 8e). Our findings suggest that depression in breast cancer patients may lead to decreased acetate production, compromised CD8[+] T cell infiltration, and ultimately, an increased risk of metastasis.

## Discussion

The association between depression and the progression of breast cancer has been well-established in previous studies[4,5]. However, the specific molecular mechanisms underlying this relationship, particularly regarding the interplay between gut microbiota, their metabolites, and the immune system, remain poorly understood. Our study contributes to the clarification of the crucial role of *Blautia* and its metabolite, acetate, in facilitating the accelerated progression of breast cancer under conditions of chronic stress. Importantly, we provide evidence that supplementation with *Blautia* and acetate significantly enhances the anti-tumor immune responses of CD8[+] T cells in the context of co-existing breast cancer and depression. Our study thus provides a mechanistic explanation for how microbial metabolic signaling, which is disrupted in cases of depression and psychological stress, affects antitumor immunity (Fig. 9).

Numerous panel studies have shown that the microbiota can influence cancer growth and metastasis. However, the identification of specific causal microbiota and their metabolites relevant to cancer

progression remains challenging. Although some studies have suggested commensals as disease biomarkers, there is still limited understanding of the precise bacterial communities and metabolic signals that explain the development of certain diseases, especially in patients with concurrent conditions such as cancer and depression[17,35,36]. To address this, a microbe-phenotype triangulation approach has been recently developed by integrating microbiome and host phenotype data to establish causal relationships between specific microbes and host phenotypes[30]. This approach has been validated as useful in eliminating inherent noise in the identification of causal microbes to certain phenotypes. Here in our study, we extended to develop a temporospatial microbe–phenotype triangulation method, which combines metagenomic and metabolomic analyses at different time points and spatial sequences. With this optimized approach, we were able to identify that the specific microbe–metabolite axis, *Blautia*-acetate, plays a causal role in depression-aggravated cancer development.

Chronic stress and depression disrupt the gut microbiota, impacting immune regulation and increasing colitis susceptibility[37]. The precise mechanisms by which microbial signals modulate the tumor-associated immune response in chronic stress remain unclear. However, microbial metabolites, specifically SCFAs, exert substantial influence on host immunity[38,39]. Our study revealed that microbiota-derived acetate might contribute to the restoration of tumor-compromised T-cell immunity. Previous studies have shown that acetate, the predominant SCFA found in circulation and tumor tissues, plays diverse roles in tuning immune homeostasis. For example, acetate was shown to support the metabolic functionality of the brain's innate immune cells and enhance their inflammatory response[40]. Additionally, the conversion of dietary fiber into acetate by the gut microbiota has been implicated in the modulation of allergic airway disease and hematopoiesis[41]. Moreover, studies have shown promising results in the utilization of acetate-producing bacteria to alleviate nonalcoholic fatty liver disease in murine models[42]. In our study, we observed a significant impact of acetate on limiting the progression of breast cancer in the presence of co-existent depression. We verified that, under conditions of depression with cancer, the decreased production of acetate may be largely explained by the reduced abundance of *Blautia*, a prominent genus within the Lachnospiraceae family. *Blautia* represents one of the most prevalent and important acetate-producing bacteria in the gut, as demonstrated by its abundance and dominance among acetate-producing strains within the family[43–46]. We also provided evidence that the supplementation of *Blautia* was able to abrogate depression and accelerate the progression of breast cancer. Since multiple microbial strains may contribute to the production of acetate, it is possible that the supplementation of other acetate-producing microbes may also be of benefit to combat depression-accelerated cancer progression. However, it is important to note that since *Blautia* represents the most dominant acetate resource and has been verified as the causal microbiota in our study, we propose that the supplementation of *Blautia*, instead of other acetate-producing

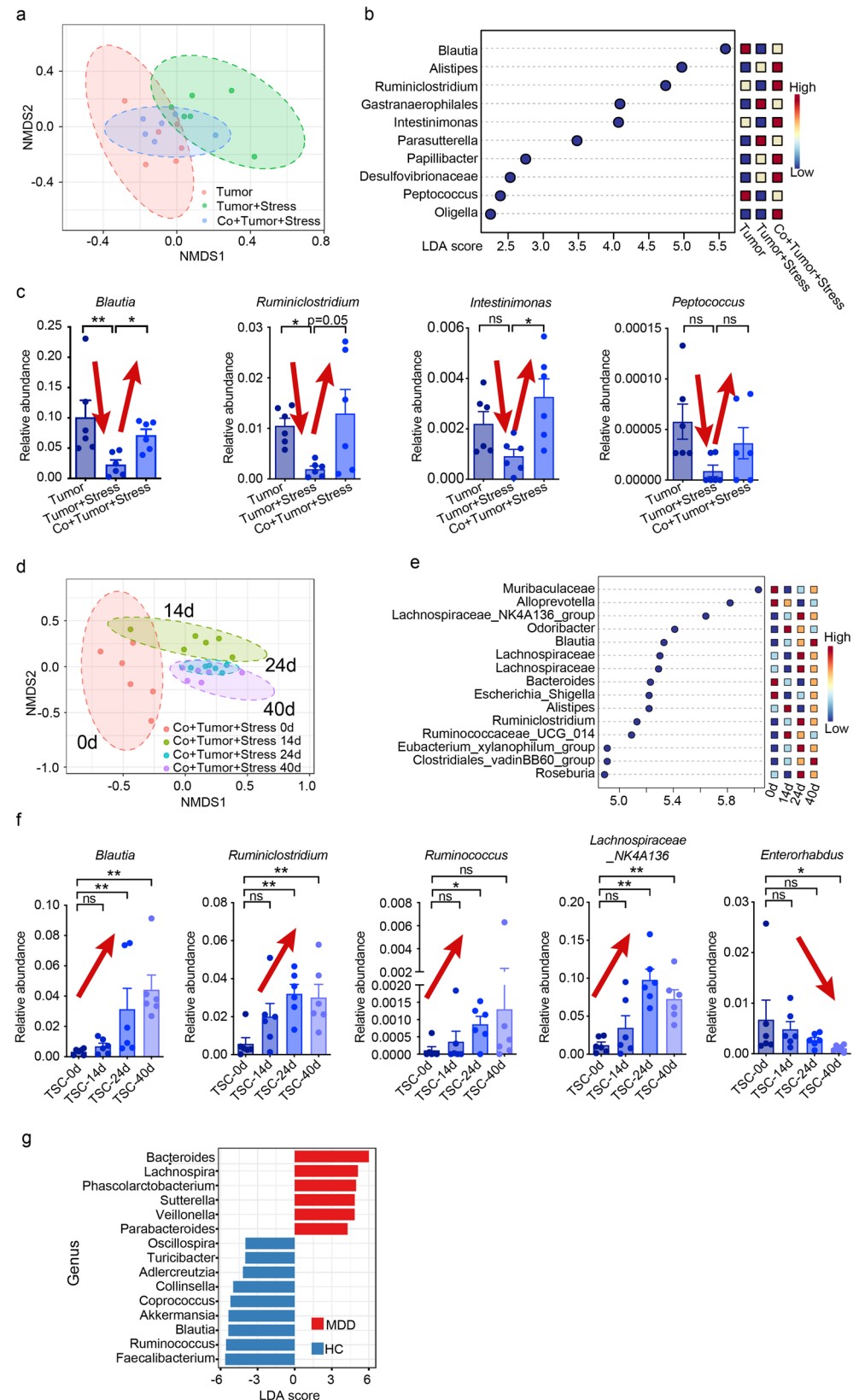

strains, may represent the most rational way in combating depression accelerated cancer progression.

Notably, our results revealed that acetate has no direct effects on CD8[+] T cells but can restore the function of impaired CD8[+] T cells when co-cultured with tumor cells. Although acetate may directly induce apoptosis in colorectal carcinoma cells at a high concentration of up to 70 mmol/L[47,48], such levels are unlikely to be reached under real pathophysiological conditions. At pathophysiologically relevant concentrations, acetate may have little direct toxic effects against tumor cells. Thus, our study, together with previous findings, indicate that acetate may have effects via interfering with the interactions between cancer and T cells rather than direct effects against either cancer or

**Fig. 3 | Temporal and spatial interlacing analysis of gut microbiota. a** Non-metric multidimensional scaling (NMDS) plot of the cecal microbiota in non-stressed, stressed, and co-housed stressed tumor mice at the end of 40 days. **b** Linear discriminant analysis (LDA) score of differentially expressed genera obtained from the LEfSe analysis of cecal microbial abundance. Genera with LDA > 2 are listed. **c** Significant changes in the relative abundance at the genus level from the cecal microbiota in the co-housing experiments. **d** NMDS plot showing time-dependent changes in the fecal microbial composition of co-housed stressed tumor mice at 0, 14, 24, and 40 days. **e** LDA score of differentially expressed genera obtained from the LEfSe analysis of fecal microbial abundance of co-housed

stressed tumor mice. Genera with LDA > 2.5 are listed. **f** Significant time-dependent changes in the relative abundance at the genus level from the fecal microbiota of co-housed stressed tumor mice (TSC). Data were presented as mean ± SEM, with $n = 6$ per group applied consistently across all panels (**a–f**). *$p < 0.05$ and **$p < 0.01$ indicate statistical significance determined by the **c** Kruskal–Wallis test and **f** two-tailed Mann–Whitney test, while "ns" indicates no significant difference. **g** LEfSe plot showing differentially abundant genera across MDD ($n = 40$) and HC ($n = 30$). Genera with Kruskal–Wallis ≤ 0.05, LDA > 2, and FDR < 0.05 are shown. Source data and exact $p$ values are provided in the Source data file.

T cells. Moreover, RNA sequencing results obtained from CD8[+] T cells co-cultured with 4T1 cells revealed a limited number of differentially expressed genes by acetate treatment. Isotopic tracing study showed that CD8[+] T cells can take up acetate and represent an important resource to feed the acetyl-CoA pool, thereby augmenting IFN-γ production and potentially enhancing the anti-tumor response. The exact molecular mechanisms underlying how acetate boosts anti-tumor T cell immunity in the tumor environment warrant comprehensive research. The present results, together with previous findings, suggest that it is important to investigate how cancer cells and T cells compete for the direct usage of acetate and how acetate functions as a signaling molecule in interfering and orchestrating interactions between cancer cells and T cells.

In order to provide a possible translational link, we have tried our best to determine whether the findings collected from animal models can implied to that of human beings. We verified that the *Blautia*-acetate axis is compromised in patients with major depressive disorder. Patients with breast cancer accompanied by depression are characterized by lower levels of serum acetate and tumor-infiltrating CD8[+] T cells compared to those without depression. One limitation of our study is the unavailability of fecal samples from breast cancer patients with and without depression, which hindered a direct assessment of how the *Blautia*-acetate axis is involved in these patients. However, results collected from animal studies, the in vitro co-culture system, and correlation analysis in human patients suggest that the compromised *Blautia*-acetate axis likely contributes to depression-promoted breast cancer progression in humans and indicate that targeting the *Blautia*-acetate axis could be a promising therapeutic approach for breast cancer patients with co-existent depression.

In summary, our study provides support for the notion that *Blautia*-produced acetate can enhance the anti-tumor response of CD8[+] T cells in depression-associated breast cancer, shedding light on a potential mechanism through which depression may promote cancer progression. Our temporospatial approach in identifying specific microbiota and metabolites relevant to pathophysiological events holds potential for broad applications. However, further comprehensive studies are needed to establish the causal link between *Blautia* and acetate in relation to depression-associated breast cancer in clinical settings and how the *Blautia*-acetate axis is orchestrated with other signals implied in depression-aggravated cancer progression. Stress hormones are known to play a role in depression-facilitated tumor development, and the gut microbiota influences the production of stress hormones through the gut-brain axis and microbial metabolites. Thus, it is likely that the stress hormones may connect to microbiota-metabolite signals in the pathological development of depression-facilitated cancer progression. Furthermore, it is crucial to further investigate the potential roles of other microbiota and their metabolites in depression-associated cancer. Future research will provide a more comprehensive understanding of the complex interactions between the gut microbiota-metabolite and neuroendocrine signals in the context of depression connected with cancer progression.

## Methods

### Ethical statement
This research complies with all relevant ethical regulations. All animal experimental procedures were conducted following the Guidance for the Care of Laboratory Animals and were approved by the Ethics Committee of Southern Medical University (Reference L2019229). Human stool samples were collected from both healthy control individuals (including 7 males and 23 females) and patients with major depressive disorder (including 12 males and 28 females) following ethical guidelines and with the approval of the Institutional Review Board of Southeast University Affiliated Zhongda Hospital (Reference 2018ZDSYLL119-P01). Human blood and breast tissue samples from female patients with breast cancer were collected with the approval of the Institutional Review Board of Nanfang Hospital (Reference NFEC-2018-038). Depression symptoms were assessed using the Hamilton depression scale (HAMD-17). Informed consent was obtained from all patients who participated in the study.

### Mice
Female Balb/c and C57BL/6 mice, aged six weeks, were obtained from Beijing Vital River Laboratory Animal Technology Co., Ltd. (Beijing, China). The mice were housed in specific pathogen-free conditions with controlled temperature (22–26 °C), humidity (45 ± 5%), and a 12-h light/dark cycle, with 4–5 mice per cage. Mice aged 7–8 weeks were used for the experiments.

### Chronic restraint stress (CRS)
Female Balb/c mice were randomly assigned to either home cage control conditions or CRS. CRS was performed according to previously reported methods[49,50]. Briefly, the mice were restrained for 6 h per day in a plastic holder that prevented them from freely moving but did not apply pressure. The behavioral effects of the stress were assessed using the tail suspension test (TST) and forced swim test (FST).

### Chronic unpredictable stress (CUS)
CUS was induced in female C57BL/6 mice following previously described methods[51–54]. Briefly, mice were subjected to two random stressors at different times of the day, including tail pinch for 15 min, inescapable foot shocks for 6 min, wet bedding with 45 °C cage tilt for 24 h, placement in a 4 °C cold room, noise in the room for 20 min, stroboscopic light for 12 h, restraint for 4 h, cage rotation for 1 h, light on overnight, and light off during the day for 3 h. Mice exposed to the CUS procedures were housed singly, while control mice were group-housed and briefly handled daily in the housing room. The behavioral effects of the stress were assessed using the FST and elevated plus maze (EPM) test.

### Breast cancer mouse model
Murine breast cancer cell lines 4T1 and E0771 were obtained from ATCC (ATCC® CRL-2539™ or ATCC® CRL-3461™) and cultured in RPMI 1640 medium supplemented with 1% penicillin–streptomycin and 10% FBS at 37 °C with 5% $CO_2$. For the subcutaneous tumor model, $2 \times 10^5$ breast cancer cells were suspended in 100 μl of 1640 medium and

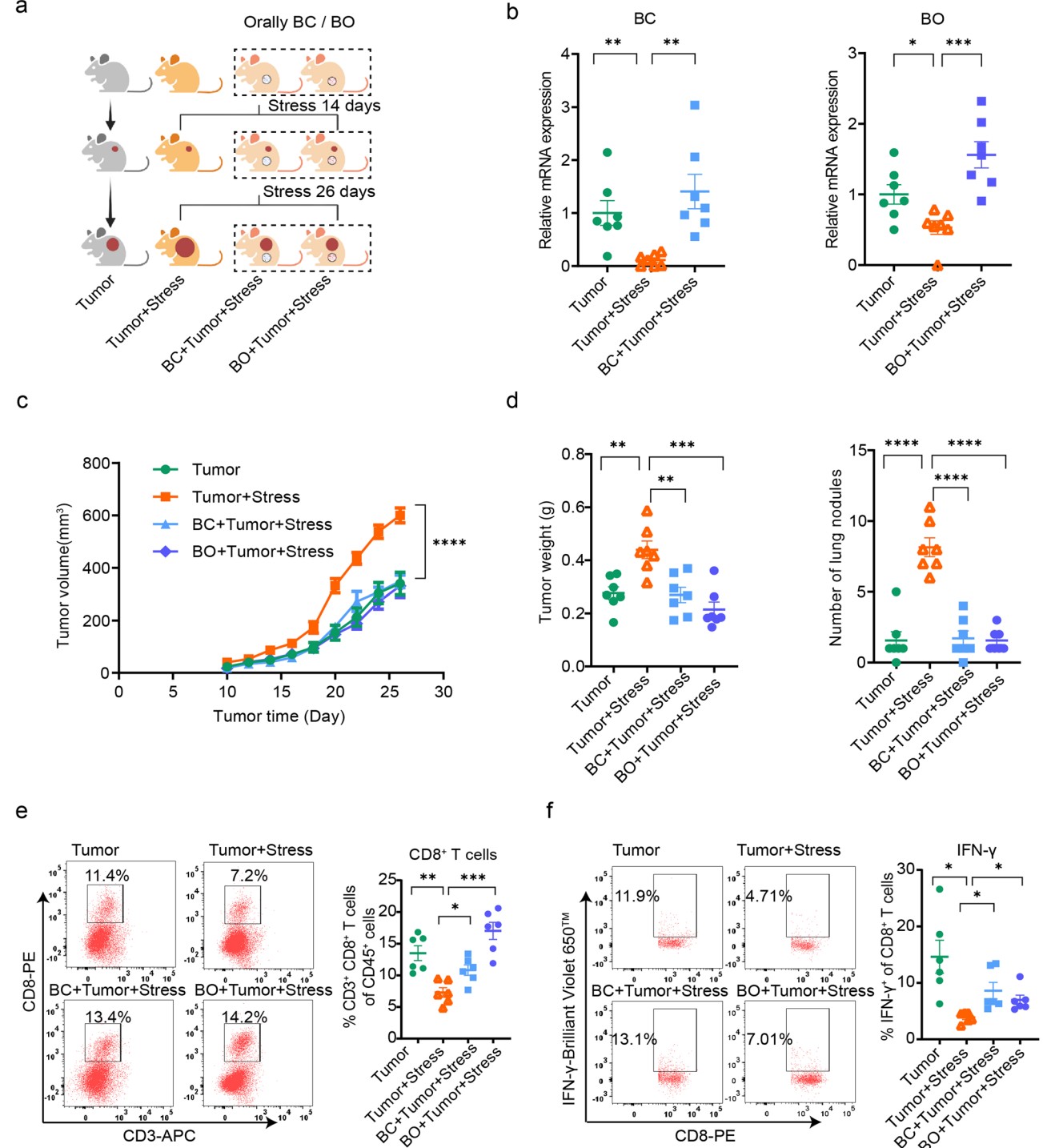

**Fig. 4 | Supplemental BC and BO eliminate promoted breast cancer progression by chronic stress. a** Schematic of the experimental design. Stressed mice were orally inoculated with or without BC/BO throughout the experiment for 40 days ($n = 7$ per group). Created with BioRender.com. **b** The mRNA expression of BC and BO after colonization for 14 days in mice was measured by qPCR ($n = 7$ per group). **c** Tumor growth volume after colonization with BC and BO ($n = 7$ per group). **d** Tumor weight and tumor burden in the lung after colonization with BC and BO ($n = 7$ per group). **e** Tumor-infiltrated CD8$^+$ T cells analyzed by flow cytometry after colonization with BC and BO ($n = 6$ per group). **f** IFN-γ abundance of tumor-infiltrated CD8$^+$ T cells after colonization with BC and BO ($n = 6$ per group). Data were presented as mean ± SEM. Statistical significance was determined by **c** two-way ANOVA followed by Sidak's multiple comparison test and **b**, **d**–**f** one-way ANOVA followed by unpaired $t$ with Welch's correction. Significance levels are denoted as *$p < 0.05$; **$p < 0.01$; ***$p < 0.001$; ****$p < 0.0001$. "ns" indicates no significant difference. Source data and exact $p$ values are provided in the Source data file.

injected into the right flank of mice[55,56]. Tumor volumes ($V$) were calculated every 2 days using the formula: $V = (L \times W^2)/2$, where $L$ is the longest diameter, and $W$ is the shortest diameter of the tumor once palpable tumors were present. Mice were sacrificed on the 26th day

after inoculation, and serum and tissue samples were collected and weighed. Maximally allowed tumor burden of 2000 mm³ was not exceeded. The lungs were fixed in Bounin's fixative for 24 h, and tumor burdens on the lungs were recorded. In some cases, prior to tumor cell

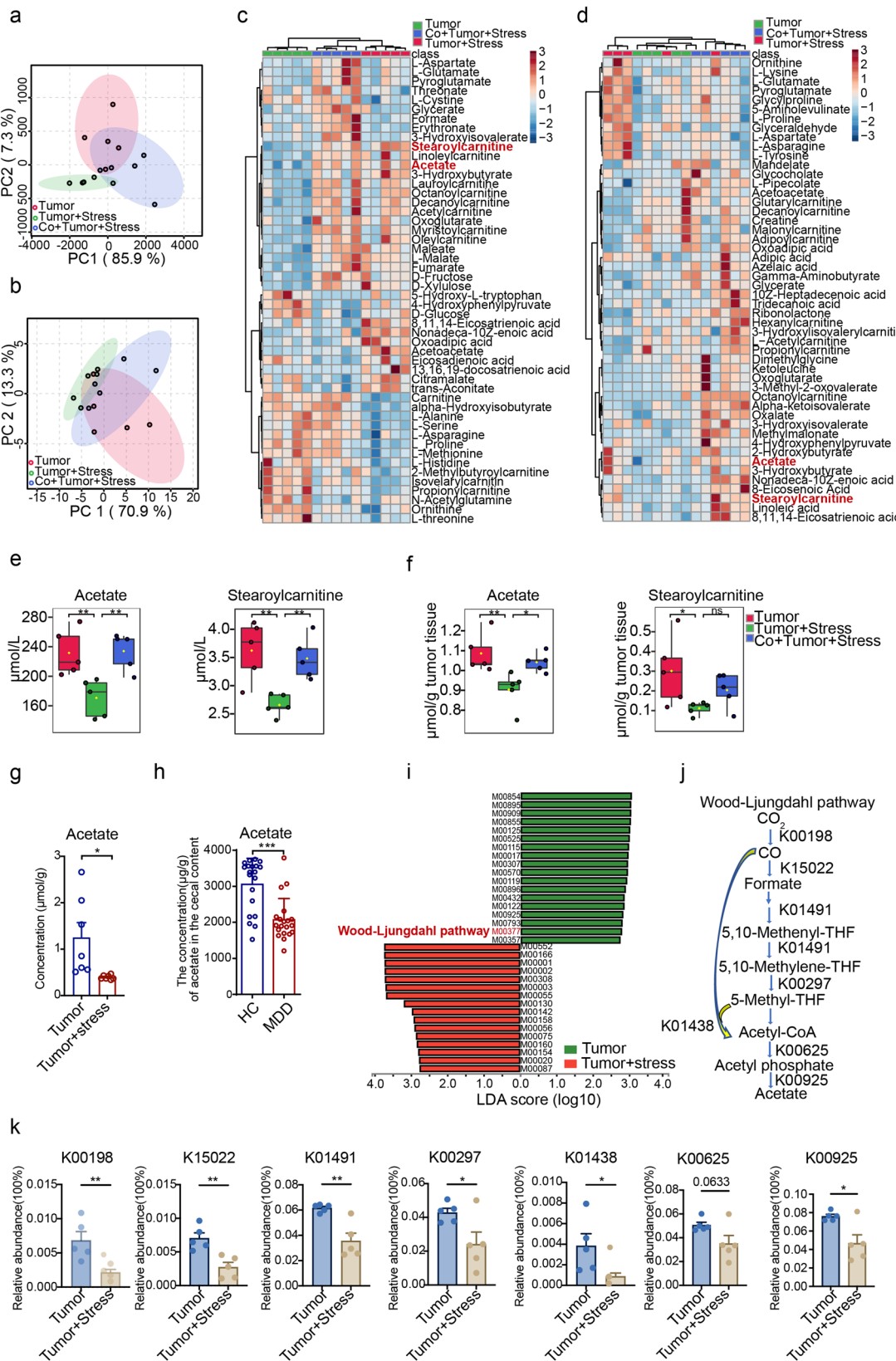

inoculation (4T1 cells for Balb/c mice or E0771 for C57BL/6 mice), mice were exposed to either CRS or CUS for a period of 2 weeks, which continued throughout the duration of the experiment. In specific cases involving Balb/c mice, a cocktail of antibiotics (0.5 g/l ampicillin, 0.5 g/l neomycin, 0.5 g/l metronidazole, 0.25 g/l vancomycin) was

administered in drinking water starting three weeks before tumor inoculation and was continued until the end of the experiment[57]. Additionally, in certain cases involving Balb/c mice, sodium acetate (NaAc) at a dose of 1 g/kg was intraperitoneally injected daily throughout the experiment[58]. Co-housing experiments were also

**Fig. 5 | Metabolomics and metagenomics analysis of microbial metabolism.** PCA score scatter based on **a** serum and **b** tumor metabolic profiling of non-stressed, stressed, and co-housed stressed tumor mice at the end of 40 days (*n* = 5 per group). Heatmap showing differential metabolic profiling in **c** serum and **d** tumor tissue among three groups (*n* = 5 per group). The concentration of acetate and stearoylcarnitine in the **e** serum and **f** tumor tissue of mice (*n* = 5 per group). The boxplot displays the distribution of concentrations for each metabolite, with the box representing the interquartile range (10th to 90th percentiles), the center line representing the median, and the whiskers representing the minimum and maximum values. **g** Concentration of acetate in the tumor tissue of tumor and stressed (CUS) tumor mice (*n* = 6 per group). **h** Contents of acetate in the feces of patients with MDD (*n* = 20) and HC (*n* = 20). **i** LEfSe analysis between tumor group and stressed tumor group in KEGG modules (*n* = 5 per group). **j** Overview of the

Wood–Ljundahl pathway: K00198: anaerobic carbon-monoxide dehydrogenase catalytic subunit; K15022: formate dehydrogenase (NADP⁺) beta subunit; K01491: methylenetetrahydrofolate dehydrogenase (NADP⁺)/methenyltetrahydrofolate cyclohydrolase; K00297: methylenetetrahydrofolate reductase (NADH); K01438: acetyl-CoA synthase; K00625: phosphate acetyltransferase; K00925: acetate kinase. (k) The relative abundance of KOs associated with the Wood–Ljundahl pathway (*n* = 5 per group). Data were presented as mean ± SEM. Statistical significance was determined by one-way ANOVA followed by the Holm–Sidak test for (**e**, **f**) or two-sided Mann–Whitney *U* test for (**k**) and unpaired two-tailed Student's *t*-test for (**g**, **h**). Significance levels are denoted as *$p < 0.05$; **$p < 0.01$; ***$p < 0.001$; "ns" indicates no significant difference. Source data and exact *p* values are provided in the Source data file.

conducted, where an equal number of stress-treated Balb/c mice with tumor inoculation and normal Balb/c mice were housed together in the same cage during the experimental procedures. For transplantation of *Blautia* species experiments, *Blautia coccoides* (BC) and *Blautia obeum* (BO) were cultured anaerobically following the instructions provided by the supplier. Bacterial cells were collected through centrifugation and suspended in PBS. Stressed Balb/c mice were orally inoculated with 200 μl of the culture suspension of BC/BO alone, which contained approximately $1 \times 10^8$ cfu (colony-forming units), per day throughout the experiment. The colonization of BC and BO in Balb/c mice was assessed using real-time quantitative PCR, targeting specific DNA sequences in the 16 S rRNA gene of the corresponding bacterial genome. In the CD8⁺ T cell depletion experiment, 100 μg of anti-CD8 antibody (clone 2.43) was intraperitoneally injected at day 2, 4, 8, 13, 17, 22, and 24 post tumor inoculation in Balb/c mice.

### Real-time quantitative PCR analysis (qPCR)
Total RNA was extracted from tumor tissue using the RNAiso Plus reagent (Takara) according to the manufacturer's instructions. The quality and quantity of the extracted RNA were assessed using methods previously described by our group[59,60]. The cDNA was synthesized from the extracted RNA, and qPCR was performed using the primers listed in Supplementary Table 1.

### Cytokine analysis
Tumor tissues were weighed and homogenized in an ice-cold PBS solution. The homogenized tissue samples were then subjected to centrifugation at 12,000×*g* at 4 °C for 15 min. The resulting supernatants were collected for further analysis. The concentrations of VEGF and IL-6 in both the tumor tissue and serum were quantified using commercially available ELISA kits, following the manufacturer's instructions.

### Flow cytometry
To analyze the immune cells infiltrating the tumor, the tumor tissue was minced on ice and digested in RPMI 1640 medium containing 3 mg/ml collagenase II/IV and 10 μg/μl DNase I at 37 °C for 60 min. The digestion was stopped by adding RPMI 1640 supplemented with 1% penicillin–streptomycin, 0.5 mM EDTA, and 10% FBS. The digested tumor tissue was then passed through a 70 μm cell strainer (BD Biosciences) to obtain a single-cell suspension. Similarly, the spleen tissue was homogenized by grinding with the plunger of a 2.5 mL syringe and filtered through a 70 μm cell strainer. To prevent non-specific binding, the cells were incubated with mouse CD16/CD32 monoclonal antibody (0.25 μg/100 μl) for 15 minutes at room temperature. Subsequently, the cells were stained with appropriate antibodies against specific antigens in a blocking buffer on ice, following the manufacturer's instructions. A viability dye was used to exclude dead cells from the analysis. Flow cytometry was performed using LSR-Fortessa instruments (BD Biosciences). The cells were gated based on viability and analyzed for various immune cell populations[25], including CD8⁺ T cells

(CD45⁺ CD3⁺ CD8⁺), T cells (CD45⁺ CD3⁺), B cells (CD45⁺ CD19⁺), NK cells (CD45⁺ CD49b⁺), macrophages (CD45⁺ MHCII⁺ CD11b⁺ CD11C⁺), neutrophils (CD45⁺ CD11b⁺ Ly6G⁺), monocytes (CD45⁺ CD11b⁺ Ly6C⁺), using the FlowJo software (Tree Star). The expression levels of IFN-γ, GZMB, and TNF-α were analyzed by flow cytometry according to the manufacturer's instructions. The gating strategy for IFN-γ analysis is depicted in Supplementary Fig. 6.

### Isolation of CD8⁺ T cells
CD8⁺ T cells were isolated using the EasySep™ Mouse CD8⁺ T Cell Isolation Kit according to the manufacturer's instructions. Spleens from female Balb/c mice were aseptically removed, mashed in 1× EasySep™ Buffer, and filtered through a 70 μm cell strainer. The resulting single-cell suspension was counted, and the concentration was adjusted to $1 \times 10^8$ cells/ml. A mixture of rat serum and antibodies (50 μl/ml) was added to the 5 ml flow tube containing the cell suspension, followed by incubation at room temperature for 10 minutes. Magnetic beads (125 μl/ml) were vortexed for 30 s, added to the tube, and incubated for 5 min. The tube was then inserted into a magnetic holder, and the cells were incubated at room temperature for 2.5 min to allow the CD8⁺ T cells to be captured by the magnetic beads. The cell suspension was poured off, collected, counted, and diluted to a final concentration of $2 \times 10^6$ cells/ml. The isolated primary CD8⁺ T cells were cultured in RPMI 1640 supplemented with 1% penicillin–streptomycin and 10% FBS at 37 °C and 5% CO₂.

### Expansion of CD8⁺ T cells and co-cultured with 4T1 cells
CD8⁺ T cells were cultured in complete RPMI 1640 culture medium supplemented with recombinant mouse IL-2 (15 ng/ml) and anti-CD28 antibody (1 μg/ml) on anti-CD3 (2 mg/ml) pre-coated plates for 12 h. Subsequently, CD8⁺ T cells ($2 \times 10^6$ cells/ml) were treated with different concentrations of NaAc, sodium propionate (2 mM), or sodium butyrate (1 mM) for 24 hours, followed by co-culturing with or without 4T1 cells ($1 \times 10^5$ cells/ml) for 48 h. Flow cytometry was used to detect the level of IFN-γ produced by CD8⁺ T cells, and death cells were labeled with propidium iodide (PI) for identification. The supernatants from the co-culture were collected, and the amount of IFN-γ was measured using an ELISA kit following the manufacturer's instructions (R&D Systems).

### RNA sequencing analysis
RNA sequencing analysis was performed on CD8⁺ T cells co-cultured with 4T1 cells treated with or without sodium acetate (5 mM). RNA isolation from purified CD8⁺ T cells was carried out using trizol reagent (Ambion) as per the manufacturer's instructions. The isolated RNA was quantified using Qubit 2.0 (Thermo Fisher Scientific) following the manufacturer's protocol[61]. The NEBNext® UltraTM RNA Library Prep Kit for Illumina® (NEB, USA) was used to prepare the libraries, which were then sequenced on a novaseq 6000 (Illumina) by the deep-sequencing facility at Novogene. The mapped reads of each sample

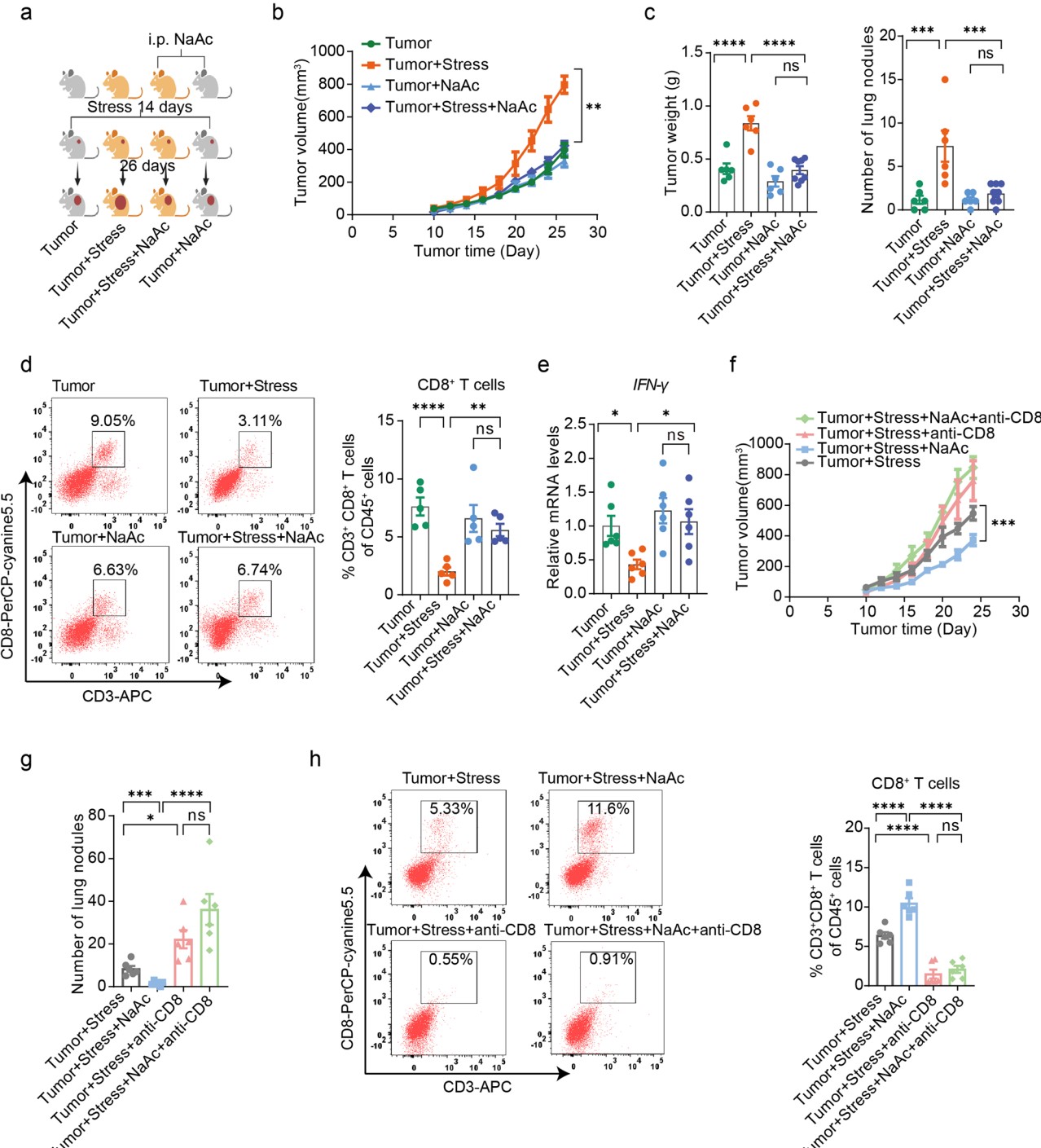

**Fig. 6 | Acetate supplementation overcomes chronic stress-promoted breast cancer progression by activating CD8⁺ T cells. a** Schematic representation of the experimental design. Mice were treated with sodium chloride (NaCl) or sodium acetate (NaAc) and then divided into two groups: one group was exposed to chronic restraint stress for 14 consecutive days before 4T1 cells inoculation and continued until the end of the experiment, and the other group was inoculated with 4T1 cells without stress exposure. The NaAc groups received NaAc throughout the experiment. Created with BioRender.com. **b** Tumor growth volume of mice treated with NaCl or NaAc and subjected to chronic restraint stress or not (Tumor group: $n = 6$, tumor + stress group: $n = 6$, tumor + NaAc group: $n = 6$, tumor + stress + NaAc group: $n = 8$). **c** Tumor weight and tumor burdens in the lungs of the mice (tumor group: $n = 6$, tumor + stress group: $n = 6$, tumor + NaAc group: $n = 6$, tumor +

stress + NaAc group: $n = 8$). **d** Tumor-infiltrating CD8⁺ T cells with or without NaAc treatment in mice ($n = 5$ per group). **e** IFN-γ expression in the tumor-infiltrating CD8⁺ T cells by qPCR ($n = 6$ per group). **f, g** Tumor growth volume and tumor burdens in supplemental NaAc-stressed tumor mice with or without depletion of CD8⁺ T cells ($n = 6$ per group). **h** Tumor-infiltrating CD8⁺ T cells in mice after NaAc treatment with or without depletion of CD8⁺ T cells ($n = 6$ per group). Data were presented as mean ± SEM. Statistical significance was determined by two-way ANOVA followed by Sidak's multiple comparisons test for (**b**, **f**), one-way ANOVA followed by Dunnett's multiple comparisons test for (**c–e**, **h**), and unpaired two-tailed Student's *t*-test for (**g**). Significance levels are denoted as *$p < 0.05$; **$p < 0.01$; ***$p < 0.001$; ****$p < 0.0001$. "ns" means no significant difference. Source data and exact $p$ values are provided in the Source data file.

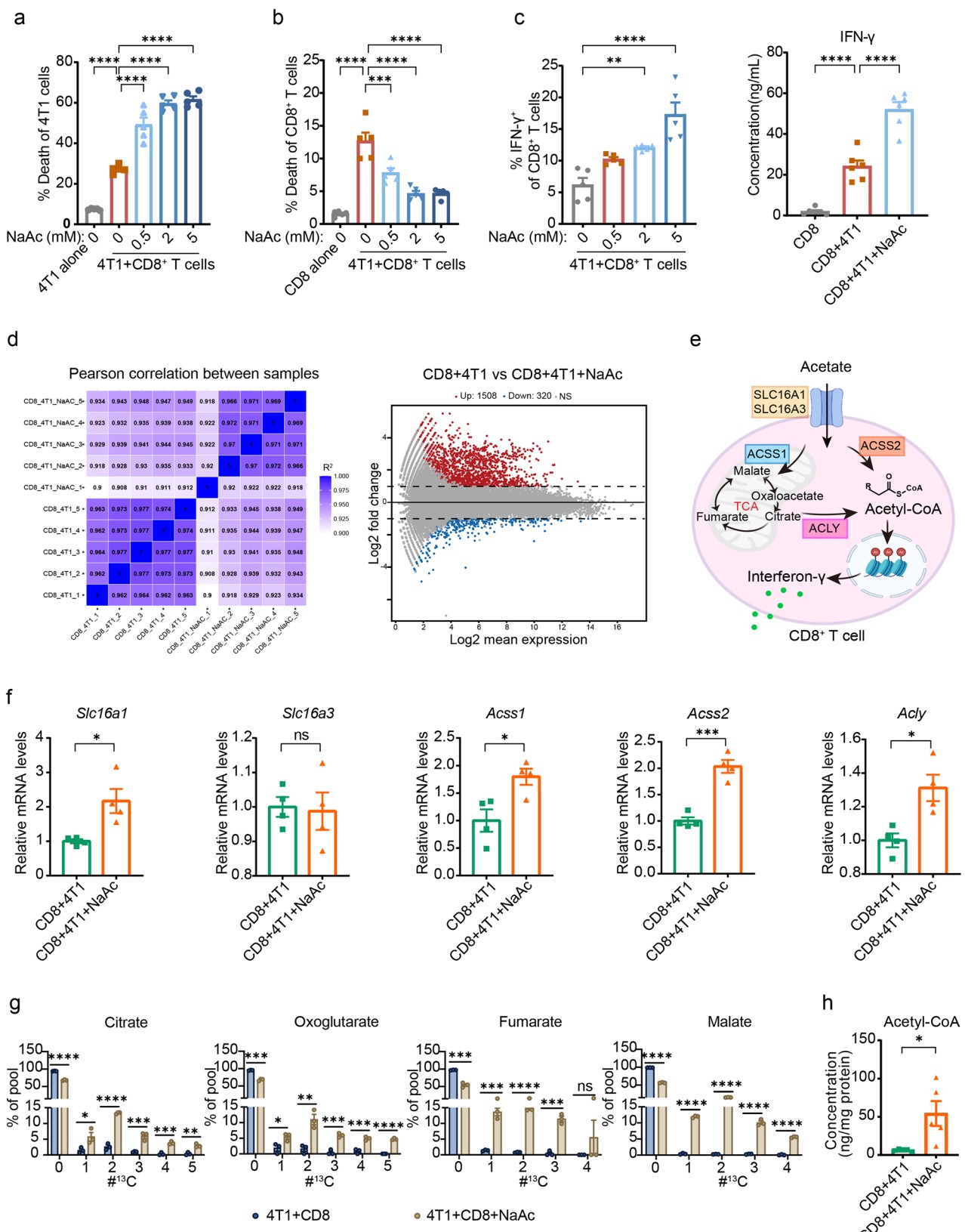

were assembled using StringTie (v1.3.3b) in a reference-based approach, and featureCounts was used to quantify the mapped reads. Differential expression analysis of two conditions/groups was performed using the DESeq2 R package (1.20.0).

**Acetyl-CoA quantification**

The concentration of acetyl-CoA in cell lysates was measured by using the CUSABIO mouse Acetyl-CoA ELISA kit, according to the manufacturer's instructions.

**Fig. 7 | Acetate enhances the activity CD8$^+$ T cells by replenishing the acetyl-CoA pool during co-culture with cancer cells. a** Percentage of 4T1 cell death after co-culture with CD8$^+$ T cells with or without NaAc ($n$ = 5 per group). **b** Percentage of CD8$^+$ T cell death after co-culture with 4T1 cells with or without NaAc ($n$ = 5 per group). **c** Quantification of IFN-γ abundance in CD8$^+$ T cells after NaAc treatment under co-culture with 4T1 cancer cells by flow cytometry ($n$ = 5) or ELISA kit ($n$ = 6). **d** Heatmap (left panel) and MA-plot (right panel) of control and acetate-exposed CD8$^+$ T cells under co-culture with 4T1 cell transcriptome ($n$ = 5 per group). **e** Pathway of acetate uptake and metabolism in CD8$^+$ T cells. Created with BioRender.com. **f** mRNA expression of the solute carrier receptors *Slc16a1*, *Slc16a3*, *Acss1*, *Acss2*, and *Acly* in control and acetate-exposed CD8$^+$ T cells under co-culture

with 4T1 cells as determined by qPCR ($n$ = 4 per group). **g** Metabolic isotopic tracing analysis of CD8$^+$ T cells under co-culture with 4T1 cells after exposure to $^{13}$C-acetate. The $x$-axis shows the number of $^{13}$C per respective metabolite. Depicted are pooled data from two independent experiments with cells from $n$ = 3 mice each. **h** Acetyl-CoA concentration in CD8$^+$ T cell lysates after co-culture with 4T1 cells with or without acetate ($n$ = 5 per group). Data were presented as mean ± SEM. Statistical significance was determined by one-way ANOVA followed by Dunnett's multiple comparisons test for (**a**–**c**) and unpaired two-tailed Student's $t$-test for (**f**–**h**). Significance levels are denoted as *$p$ < 0.05; **$p$ < 0.01; ***$p$ < 0.001; ****$p$ < 0.0001. "ns" means no significant difference. Source data and exact $p$ values are provided in the Source data file.

## Immunohistochemistry

Immunohistochemistry was employed to analyze protein expression in breast cancer tissues from patients with or without depression. The tissue sections were incubated overnight with primary antibodies against CD8 at 4 °C. Mayer's hematoxylin was used for nuclear counterstaining. CD8-positive cells within the tumors were quantified by two independent pathologists through manual counting in five high-powered fields (HPFs).

## Metabonomic analysis for serum and tumor samples

The UPLC-MS/MS system (ACQUITY UPLC-Xevo TQ-S) was utilized for the absolute quantification of serum, tumor, and stool samples, as previously described[62,63]. For serum samples, 20 μl aliquots were mixed with 120 μl internal standard solution, centrifuged at 13,500$g$ and 4 °C for 10 min, and 30 μl of the supernatant was transferred to a 96-well plate for derivatization. For tumor tissue and stool samples, approximately 10 mg were weighed and homogenized with 20 μl water, extracted with 150 μl cold methanol with the internal standard mix, and centrifuged at 13,500$g$ and 4 °C for 10 min. In total, 30 μl of the supernatant was transferred to a 96-well plate for derivatization. Chromatographic separations were performed using an ACQUITY BEH C18 column (1.7 μm, 100 mm × 2.1 mm internal dimensions) (Waters, Milford, MA). The mass spectrometer was operated in negative mode with a 2.0-kV capillary voltage and in positive mode with a 1.5-kV capillary voltage. The source and desolvation temperatures were set at 150 and 550 °C, respectively. The TMBQ software (v1.0, HMI, Shenzhen, Guangdong, China) was used for peak integration, calibration, and quantification of each metabolite from the targeted raw data. The current version of TMBQ (GNU GPL.V3 license) is implemented with Java and R and is freely available at http://119.136.25.134:9011.

## Gut microbiota profiling by 16 S rRNA sequencing of cecal and stool sample

The 16 S rRNA sequencing of cecal/fecal/stool samples were conducted with slight modifications following our previously described protocol[49,64]. Bacterial DNA was extracted from cecal and stool samples using the E.Z.N.A.® soil DNA Kit (Omega Bio-tek), and the DNA concentration and purity were assessed with the NanoDrop 2000 UV-vis spectrophotometer. The hypervariable region V3-V4 of the bacterial 16 S rRNA gene was amplified using primer pairs 338 F (5'-ACTCC-TACGGGAGGCAGCAG-3') and 806 R (5'- GGACTACHVGGGTWTCTAAT-3') by the ABI GeneAmp® 9700 PCR thermocycler. The purified amplicons were pooled in equimolar amounts and subjected to paired-end sequencing on an Illumina MiSeq PE300 platform/NovaSeq PE250 platform (Illumina, San Diego, USA) according to the standard protocols by Majorbio Bio-Pharm Technology Co. Ltd. (Shanghai, China). Raw fastq files were demultiplexed and quality-filtered using QIIME, and the sequences were matched against a high-quality 16 S rRNA sequence from the Green Genes database after trimming the primer, barcode, and chimeras. Operational taxonomic units (OTUs) were generated at a 97% similarity cutoff using UPARSE (version 7.1; http://drive5.com/uparse/), and chimeric sequences were detected and eliminated using UCHIME. Principal coordinates analysis (PCoA), non-metric multidimensional

scaling (NMDS), and LEfSe analysis were conducted using Microbiome Analyst (https://www.microbiomeanalyst.ca/). We performed the 16 S rRNA sequencing and data analysis on fecal samples, as previously reported, with minor adjustments. For LEfSe analysis, we presented the genus with Kruskal–Wallis ≤ 0.05, LDA > 2, and FDR < 0.05.

## Shotgun metagenomics

The concentration and purity of the extracted DNA from non-stressed and stressed tumor mice were determined using TBS-380 and Nano-Drop2000, respectively. The quality of the DNA extract was evaluated on a 1% agarose gel. The DNA was then fragmented to an average size of approximately 300 bp using Covaris M220 (Gene Company Limited, China) for paired-end library construction. The paired-end library was constructed using NEXTFLEX Rapid DNA-Seq (Bioo Scientific). Adapters containing the full complement of sequencing primer hybridization sites were ligated to the blunt end of the fragments. Paired-end sequencing was then carried out on the Illumina HiSeq4000 platform (Illumina Inc., San Diego, CA, United States) located at Majorbio Bio-Pharm Technology Co., Ltd. (Shanghai, China). The HiSeq 3000/4000 PE Cluster Kit and HiSeq 3000/4000 SBS Kit were utilized according to the manufacturer's instructions. The raw sequencing data were evaluated using FastQC, and Trimmomatic filtering was employed to obtain relatively accurate and valid data. Open reading frames (ORFs) were predicted for each assembled contig using Prodigal. Predicted ORFs with lengths of 100 bp or more were searched and translated into amino acid sequences using NCBI translation tables. Representative sequences from the nonredundant gene catalog were aligned to the NCBI NR database using BLASTP (version 2.2.28 + ) with an e-value cutoff of 1e-5 for taxonomic annotation. To target the Kyoto Encyclopedia of Genes and Genomes database (http://www.genome.jp/keeg/), BLASTP (Version 2.2.28+) was utilized with an e-value cutoff of 1e-5.

## Metabolic isotopic tracing analysis of CD8$^+$ T cells after exposure to $^{13}$C-acetate

CD8$^+$ T cells were incubated with $^{13}$C-acetate (5 mM) under co-culture conditions with 4T1 cells. To obtain intracellular metabolites, CD8$^+$ T cells were washed twice with pre-cooled PBS, and their metabolism was halted by adding 80% ice-cold methanol. Subsequently, a pre-cooled mixture of 80% methanol (1 mL) containing para-chlorophenylalanine (1 μM) as an internal standard was added to each sample, which was then incubated on ice for 10 min. The samples were then centrifuged at 4 °C for 15 min at 23,000×$g$, and the supernatants were collected and evaporated under vacuum prior to LC−MS analysis. The compounds were separated using an Agilent 1290 Infinity II UPLC system with an XBridge BEH Amide column (4.6 mm × 100 mm, 3.5 μm; Waters) and detected using an Agilent 6546 Q-TOF mass spectrometer equipped with a dual jet stream electrospray ionization source operating under extended dynamic range (EDR 1100 m/z) in negative ionization mode. The mobile phase consisted of two eluents, eluent A (15 mM ammonium acetate and 15 mM ammonium hydroxide in a 95:5 water/acetonitrile mixture, pH = 9) and eluent B (acetonitrile). Three microlitres of the samples were injected via an autosampler into

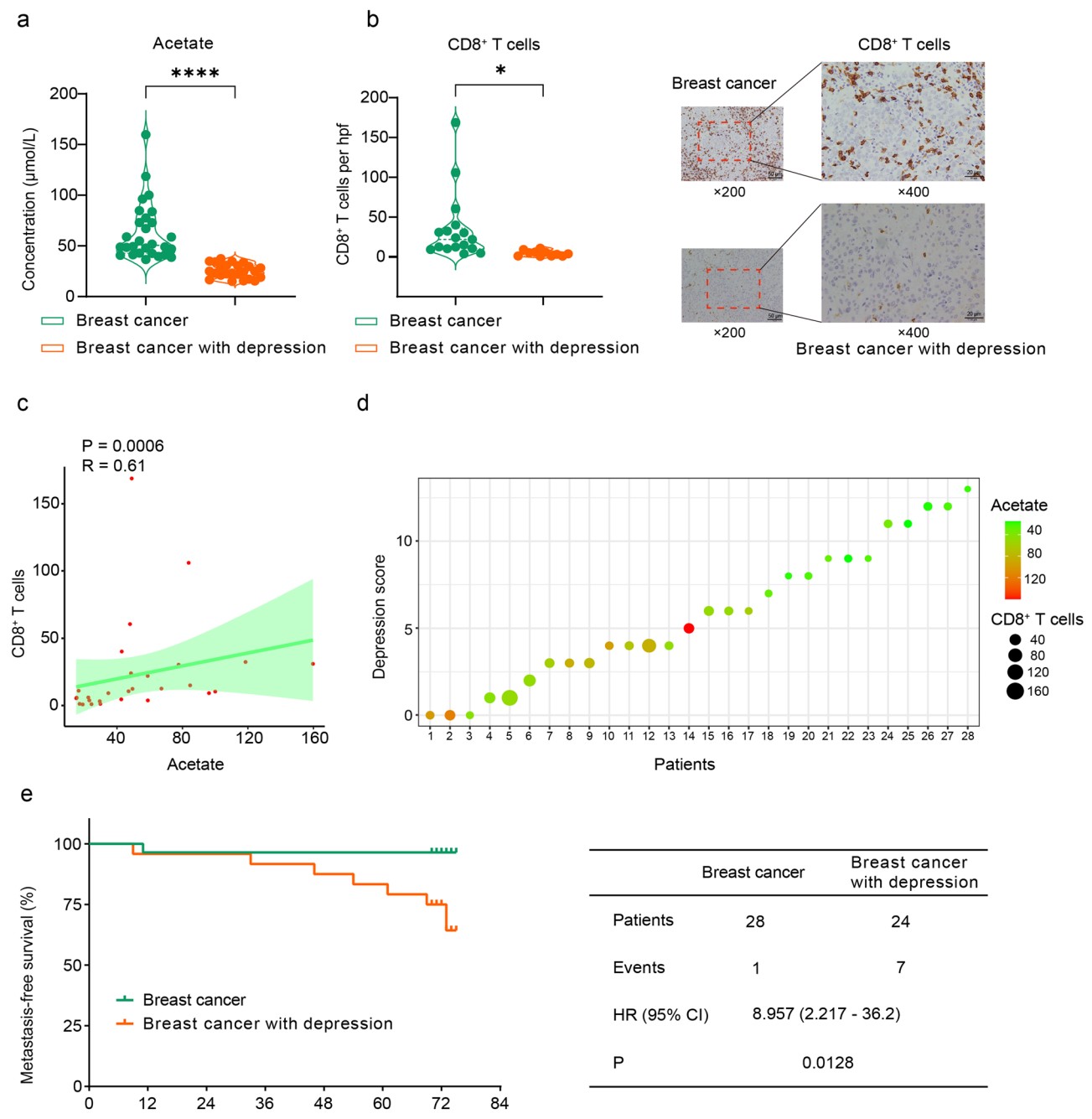

**Fig. 8 | Acetate levels correlate with CD8⁺ T cells in breast cancer patients.**
**a** Serum acetate levels in breast cancer patients with and without comorbid depression ($n = 28$ and $n = 24$, respectively). **b** Immunohistochemical staining of CD8⁺ T cells on breast cancer tissue samples obtained from patients with and without comorbid depression ($n = 17$ and $n = 11$, respectively). The scale bars in the images represent 50 μm and 20 μm, respectively. **c** Spearman correlation analysis between serum acetate levels and the count of tumor-infiltrating CD8⁺ T cells ($n = 28$). The correlation was accessed by Spearman's rank correlation coefficient ($R = 0.61$). The regression line is green, and the shading indicates the 95%

confidence interval. **d** The relationship among serum acetate levels, count of tumor-infiltrating CD8⁺ T cells, and the depression score of breast cancer patients. Each sample ($n = 28$) was represented by a dot, with the dot's size indicating the CD8⁺ T cell count and its color representing the serum acetate levels.
**e** Kaplan–Meier metastasis-free survival curves for breast cancer patients with and without comorbid depression ($n = 28$ and $n = 24$, respectively). Statistical significance was determined by unpaired two-tailed Student's $t$-test. Significance levels are denoted as \*$p < 0.05$; \*\*$p < 0.01$; \*\*\*$p < 0.001$; \*\*\*\*$p < 0.0001$. Source data and exact $p$ values are provided in the Source data file.

the mobile phase, and chromatographic separation was achieved at a flow rate of 0.4 ml min⁻¹. The gradient was as follows: 0 min, 85% B; 1 min, 85% B; 12 min, 30% B; 14 min, 2% B; 15 min, 2% B; 15.5 min, 85% B; 30 min, 85% B. Agilent MassHunter Pathways to PCDL software was used to construct a database of compounds related to the tricarboxylic acid (TCA) cycle. Metabolite retention times were added to a custom PCDL, which was then used as a database to perform bulk isotopic

heteroconiosis extraction in Profinder. The results generated by Profinder were exported and visualized using Omix Premium software.

## Statistical analyses

The data analysis was conducted using GraphPad Prism 7.0 software. Unpaired two-tailed Student's $t$-test was used for comparisons between two groups, while one-way ANOVA was used for multiple

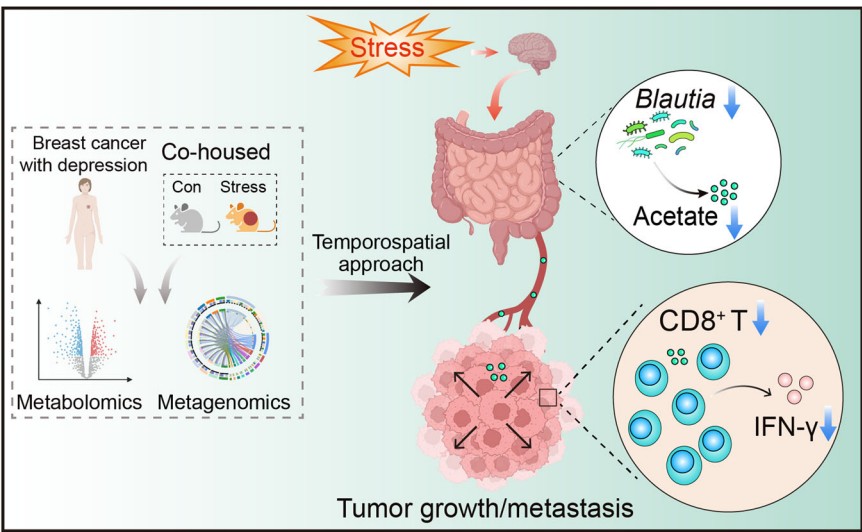

**Fig. 9 | Deficiency in *Blautia*-produced acetate underlies breast cancer progression promoted by chronic stress.** In the present study, using an effective approach that combined temporospatial mapping of gut microbiota and its metabolites with metabolomics and metagenomics, we reveal that chronic stress results in reduced numbers of the causal microbe (*Blautia*) and its metabolite (acetate). These deficiencies may underlie chronic stress-promoting breast cancer progression by regulating the reduction of the number and function of CD8[+] T cells in the tumor microenvironment. This figure presents a schematic representation of the proposed mechanism, illustrating the role of *Blautia*-produced acetate in regulating CD8[+] T cells and ultimately impacting tumor growth in the conditions of co-occurrence of depression and breast cancer. Created with BioRender.com.

group comparisons unless otherwise indicated. The data were presented as mean ± SEM and statistical significance was denoted in figures using asterisks as follows: $*p < 0.05$, $**p < 0.01$, $***p < 0.001$, and $****p < 0.0001$.

### Reporting summary
Further information on research design is available in the Nature Portfolio Reporting Summary linked to this article.

## Data availability
The 16 S rRNA and metagenomic sequencing datasets are available in the NCBI SRA database under the accession numbers PRJNA912999 and PRJNA978792, respectively. Additionally, the RNA sequencing data have been deposited in GEO with the subseries code GSE231961. Source data are provided in this paper. The remaining data are available within the Article, Supplementary Information, or Source data file. Source data are provided in this paper.

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

## Acknowledgements

This study was financially supported by grants from the National Key Research and Development Program of China (grant 2021YFA1301300 to H.H.); the National Natural Science Foundation of China (grants 81930109 to H.H.; 82074110 and 82274193 to L.Y.; 82104280 to Y.H.); the Open Project of State Key Laboratory of Natural Medicines (grant SKLNMKF202209 to L.Y.); the Project of Shenzhen Science and Technology (grant JCYJ20210324134809025 to Y.H.); China Postdoctoral Science Foundation (grant 2021M693512 to Y.H.).

## Author contributions

L.Y., W.H. and Y.H. conducted the experiments with assistance from R.Y. and X.Z. and L.Y. and Y.H. analyzed data. H.W., Q.Z., Q.F. and G.Y. collected and analyzed the human samples. H.H. supervised the project; L.Y., Y.H. and H.H. wrote and revised the paper.

## Competing interests

The authors declare no competing interests.
