## [Peer Review File · Nature Communications]

Repressed Blautia-acetate immunological axis underlies breast cancer progression promoted by chronic stressREVIEWER COMMENTS

Reviewer #1 (Remarks to the Author): with expertise in breast cancer, stress, cancer immunology

In their article "Repressed Blautia-acetate immunological axis underlies chronic stress promoted breast cancer progression" the authors focus on understanding how depression, and immunosuppression associated with it, promotes tumor growth. They focus on the microbiome and metabolites which are critically involved in immune responses. Female BALB/c mice are subjected to daily restraint stress to generate a model of depression and the growth of 4T1 mammary tumors is analyzed.

The authors convincingly show that stressed, tumor bearing mice have reduced levels of Blautia and its metabolite acetate. Further, treatment of stressed, tumor bearing mice with either strains of Blautia or acetate can negate the tumor promoting effects of chronic stress. Additionally, these levels correlate with tumor infiltration by CD8+ T cells. Using patient material, they show that levels of acetate are significantly reduced in BreCa patients with depression compared to non-depressed patients.

Overall, this is a "proof of principal" paper showing that a specific strain of bacteria and one of its metabolites is reduced by stress and can be used therapeutically to overcome the tumor promoting effects of stress.

Many questions remain, including whether Blautia plays the same role in other strains of mice and tumor models. Also, it remains to be clarified how stress itself modulates the microbiome.

Abstract:

The abstract states that comparison of BreCa patients indicated that depressed patients had reduced abundance of Blautia and acetate compared to non-depressed patients, but does not say whether the tumor sizes were larger or whether the prognosis or outcome is worse? If this information is available, please add it to the manuscript and at least, the Discussion.

Use of the term “regression”. Regression implies that the tumors were large and became smaller with treatment. A different word would better convey the observation that the tumor-growth promoting effects of stress could be overcome by treating stressed, tumor bearing mice with Blautia or acetate. “Negates” is a more accurate and this wording should be changed in multiple places. The authors have used “eliminates” in some instances, and that is also a good choice.

Throughout the manuscript, there are places where the text is written “Blautia-produced acetate underlies chronic stress promoted breast cancer progression”. This implies that there is a direct relationship and that Blautia is mediating the progression when the authors mean the opposite- that Blautia prevents stress induced progression. Please revise similar to how it is phrased in line 295:

“Chronic stress leads to a gradual loss of Blautia and thereby reduced production of its metabolite acetate, which may underlie the reduced infiltration and impaired function of CD8+ T cells in the tumor, thereby promoting breast cancer growth and metastasis.”

Co-housing experiments: Please explain the rationale for this and the experimental design more clearly. I presume the co-housing is to make sure the microbiomes are the same? Was this assessed?

IL-6 and vegfc levels are reported, but there is not explanation of why and they aren't included in the conclusions. Please expand on the significance of these factors, or omit them.

Line 127-128. Suggested re-wording...

“Results showed that the accelerated tumor growth-promoting effects of stress of breast cancer was regressed were lost when stressed mice were in co-housed with non-stressed mice, stress-treated tumor mice (Fig. 2h-2i), along with decreased expression levels of IL-6 and vegfc (Fig. S2d-2e)”

Line 141

“microbe-phenotype triangulation”- Could this be briefly explained?

Figures, the font is too small and extremely hard to read (6pt in figures as printed)

Line 201: In addition, the supplemented strains of Blautia (BlautiaBlautia coccoides and BlautiaBlautia obeum) that significantly regressed negated cancer progression by chronic stress promoted cancer progression were also able to produce large amounts of acetate (Fig. S4f). There are several places where Blautia is misspelled including line 28 in the abstract.

Fig 6, legend title

“Acetate supplementation overcomes chronic stress-promoted breast cancer progression by activating CD8+ T cells

Fig 7- define hpf

Fig 8- Title should read Deficiency in Blautia-produced acetate underlies chronic stress promoted breast cancer progression. In present study, using an effective approach which combined temporospatial mapping of gut microbiota and its metabolites with metabolomics and metagenomics, we revealed that chronic stress results in reduced numbers of the causal microbe (Blautia) and its metabolite (acetate). These deficiencies may underlie chronic stress promoted breast cancer progression by regulating reducing number and function of CD8+ T cells function in the tumor microenvironment.

Reviewer #2 (Remarks to the Author): with expertise in breast cancer, microbiota

In the manuscript of ‘Repressed Blautia-acetate immunological axis underlies chronic stress promoted breast cancer progression’, Ye et al. addressed an interesting question of how chronic stress impinges on the gut microbiome and whether this stress associated gut microbiome alteration has any influence on the tumor progression. They reported that chronic stress can promote tumor growth and lung metastasis, and this tumor promoting

effect was dependent on the gut microbiome, as depletion of gut microbiome abolished the stress mediated effect. After analysis of the gut microbiome composition, they pinpointed that *Blautia* was the key bacteria in the gut microbiota mediating stress associated tumor growth and metastasis. Mechanistically, they reported that *Blautia*'s metabolite acetate can promote CD8+ T cells infiltration and activation, therefore reduced *Blautia*-acetate axis accelerate the progression of breast cancer by chronic stress. In addition, they found acetate was correlated with CD8+ T cells in breast cancer patients. The scientific question they asked is an interesting one and is important for the cancer field. However, I am not fully convinced due to some experimental flaws and some inconsistencies. My major criticisms are listed below:

1. Figure 1g, 2K, Please show the gating strategy. Contour plot is not good to show the positive population, please show the dot plot, also show the negative controls. From the contour plot, the IFN γ staining does not seem to work.
2. *Blautia* might not be the only acetate producing bacteria in the gut, why other acetate producing bacteria can not compensate the effect of *Blautia* reduction?
3. The effect of ABX treatment on gut microbiome was not quantified. It is hard to judge whether it is complete abolishment of gut microbiome or it is just dysbiosis.
4. Figure 3a-c and 3d-f are both co-housing experiments, however, the bacteria dynamics are quite different, *Ruminiclostridium* showed significant change in one experiment but not the other. *Lachnospiraceae* and *ruminococcus* showed up in one but not the other. How variable is this experiment?
5. In Figure S3b, tumor graft seems to increase the *Blautia* abundance, which is supposed to enhance T cell activity, would it be the opposite? In addition, the *blautia* level in Tumor+stress group is reduced to a similar level to wt control group, why co-housing of the two mice with similar *Blautia* level can rescue the Tumor-stress *blautia* level?
6. From Figure 7a, the concentration of acetate in breast cancer patients' serum is 25-160 μ g/ml, and mouse serum and tumor tissue showed similar concentration, while in Figure 6, NaAc was given 1g/kg intraperitoneally per day, which is way more than the physiological concentration. In addition, Figure 5f showed acetate level in tumor tissue, stress only reduced acetate level by ~20% with big variation, and 20% of acetate alteration can cause striking phenotype, why need to administer such a high level of NaAc to rescue the

phenotype?

7. According to the model, Blautia- acetate-T cell axis mediated chronic stress associated breast cancer progression. Therefore, Blautia and acetate should be the downstream of stress. However, ABX treatment, Blautia inoculation and acetate supplement were performed before chronic stress exposure and tumor inoculation. Do these treatments influence tumor initiation or tumor progression?

8. Figure 7b. The high-resolution pictures should be provided. IHC staining of CD8+ T cell in depressed breast cancer patients' tissues seems the adjacent normal tissue. and the breast cancer types should be recorded in a table.

Minor concerns are:

1. There are some inconsistencies in tumor growth: Figure 1b, Figure 4c and Figure 6b, the tumor started to growth from day 10 while in Figure 2b, it was around day 7. Moreover, at day 26, why the tumor volume of tumor, tumor+stress group are different in Figure 1b, Figure 2b, Figure 4c and Figure 6b. Is the stress model stable?

2. There are some typos, line 28, 173, 177, 201, 205, 742 "Blatua", should be "Blautia"; line 787 "corrected" should be "correlated".

3. The phylum and genus distribution of gut microbiota in each condition for each individual mouse should be shown in the supplementary figures for better judgement of the data quality.

4. The color label in Figure 3b for Peoptococcus is not consistent with Figure 3C bar plot.

Reviewer #3 (Remarks to the Author): with expertise in metagenomics, metabolomics

Remarks to the Author:

In this study, titled " Repressed Blautia-acetate immunological axis underlies chronic stress promoted breast cancer progression" by Ye et al, the authors used a navigational strategy called "microbe-phenotype triangulation" to pinpoint casual protective microbe that shape host physiology and disease susceptibility. They identified Blautia and its metabolite acetate as protective factors against breast cancer progression under chronic stress. The underlying mechanism is to promote anti-tumor immune response by restoring the function of tumor

infiltrating CD8+ T cells.

The study is well described and structured. Find below major comments which, if addressed, would significantly strengthen the manuscript.

Introduction

It is recommended to first present global epidemiological data on breast cancer, including incidence, mortality, etc. Based on previous studies, the authors should then list the pathogenesis of breast cancer and then describe the impact of depression as one of the factors on the development of breast cancer. Finally, the Blautia-acetate immunological axis was introduced, focusing on its role in possible influence in the development of breast cancer.

Methods

1.As far as I know the common mouse models of depression include stress modeling, drug modeling, surgical modeling and genetic modeling. Chronic restraint stress mice can partially describe a state of psychological stress but are prone to adaptation. Whether the use of other experimental animal models could better validate the phenomenon found by the authors ?

2.I have seen only one breast cancer murine cell line used by the authors, and I think at least two different cell lines are needed for validation.

Results

1.The description of the results needs to be concise and no specific analysis is needed here, e.g., the body weight of mice in group A is higher than that in group B, and the interferon γ of mice in group A is lower than that in group B. Detailed description of the treatment of mice in the method section, and only a brief indication in the results is needed.

2.In Figure 5e&f the authors showed acetate and stearyl carnitine in serum and tumor samples represented the most significantly changed metabolites by co-housing. The authors are requested to specifically label these two metabolites in Fig 5c and d to allow us to better see the variation in their levels in different groups.

3.Is acetate produced exclusively by Blautia, or do other bacteria also metabolize acetate?

How important is the role of Blautia in this, and is it possible to knock out the enzyme in which Blautia metabolizes acetate, or is it possible to specifically delete the bacteria?

4. The authors' findings suggested that acetate may have no direct effect on cancer cells and CD8+ T cells, but may prevent cancer-induced impairment of CD8+ T cells, thereby enhancing anti-tumor immunity. This is a very interesting phenomenon, can the author explain the mechanism or the possible reasons for it?

5. I think it is a great pity that stool samples from patients with breast cancer combined with depression were not collected for validation in this study.

Discussion

The article focuses on the effect of Blautia-acetate immunological axis in breast cancer, which requires a detailed analysis of its underlying mechanism. Firstly, analyze the effect of Blautia-acetate immunological axis in other diseases, then analyze the effect of Blautia-acetate immunological axis in the context of the results of this study, confirm where your results agree or disagree with others, and analyze the reasons for the disagreement.

Reviewer #4 (Remarks to the Author): with expertise in cancer immunology

In this manuscript, Ye L. and colleagues postulated that the Blautia-generated SCFA acetate suppresses stress-promoted breast cancer progression by activating T cell-mediated anti-tumor immunity. The authors show increased frequency of tumor-infiltrated IFN- γ -expressing CD8+ T cells following treatment of mice with acetate. Moreover, the acetate levels are positively associated with tumor-infiltrated T cells in breast cancer patients. While some of the results are potentially very interesting, there are several issues that should be thoroughly addressed.

Major concern:

The essential part of the paper is the acetate-induced tumor regression of stress-promoted tumor progression. It is important to analyze the effects of acetate on the survival and cytotoxic capacity of CTLs in the co-culture with 4T1 cancer cells in more detail. In contrast to butyrate, acetate appears not to be able to directly affect the functionality of CTLs.

Further functional experiments are required to understand mechanisms underlying acetate-

mediated effects. Metabolic flux analysis of co-cultured CTLs and 4T1 cells using ¹³C-labeled sodium acetate is needed to understand metabolic profile of cells following acetate treatment (high glucose vs. low glucose). The authors also should explore the relation between acetate treatment of co-cultures (as compared to CTLs without 4T1 cells) and possible alterations in glycolysis and OXPHOS. Are glucose-restricted CTLs in the co-culture more or less responsive to acetate-mediated effects as compared to normal CTLs? Some papers have shown a direct effect of acetate on CTLs (Luu Maik et al., *Sci. Rep.*, 2018), and acetate is even incorporated into histones of CTLs (Qiu Jing et al., *Cell Reports*, 2019). The authors should critically discuss the established literature (e.g. the paper: Systemic short chain fatty acids limit antitumor effect of CTLA-4 blockade in hosts with cancer, *Nature Comms.*, 2020), and they might also provide the data for other SCFAs in the co-cultures (CTLs and 4T1 cancer cells).

The methods used in this study are sound. The authors have included both, in vivo experiments and translational data. However, there is too much focus on a possible link between depression in breast cancer patients and *Blautia*-acetate axis. In the experimental model, the chronic stress enhances breast cancer progression. In contrast, in humans, diagnosis of cancer can lead to serious depression. In terms of mechanistic understanding, the data from the Fig. 7 are overinterpreted. For the Fig. 7b (IHC), authors analyzed a cohort of only 5 breast cancer patients without depression and 8 depressed cancer patients. I am not sure if the Fig. 7 really supports the conclusions and claims. For their primary statements, authors should use a larger group of patients and should also more focus on the mechanistic part of the study in the murine experimental model of the stress-promoted cancer progression. It would be important to generate a mutant strain of *Blautia coccoides* that is not able to produce acetate to analyze *Blautia*-acetate-mediated effects in vivo. It would be also important to establish a mechanistic link between *Blautia*-acetate deficiency and stress hormones that are known to promote tumorigenesis in order to strengthen the current claims.

Of note, acetate is most abundant SCFA, which is produced by most of the enteric bacteria. Why should only acetate generated by *Blautia*, and not that produced by other bacteria, play an important role?

The Fig. 2 may suggest that reduced acetate levels would have opposite effect as compared to other figures. It is known that vancomycin exposure leads to depletion of acetate-producing bacteria in the gut and that mice treated with antibiotics produce less acetate. This figure suggests that gut microbiota drives the stress-associated breast cancer progression. How can the authors explain this observation?

Collectively, the authors could try to explore more mechanistic findings in mice, instead of strong focus on patients in the abstract and discussion. Does the data generated with samples from depressed breast cancer patients support all conclusions and claims? It would be of great benefit if the authors can test their hypothesis in another and larger cohort of depressed cancer patients (across various cancer types).

Minor points:

Line 787 (Figure legend of the Fig 7.) the word “corrected” should be replaced by the word “correlated”.

In the Figs. 3c and 3f, the authors use the terminology: “Blautia / relative expression”. I do not think that bacteria can be “expressed”? I would suggest the words: “relative abundance”.

Response to Reviewers

REVIEWER COMMENTS

Reviewer #1 (Remarks to the Author): with expertise in breast cancer, stress, cancer immunology

In their article "Repressed *Blautia*-acetate immunological axis underlies chronic stress promoted breast cancer progression" the authors focus on understanding how depression, and immunosuppression associated with it, promotes tumor growth. They focus on the microbiome and metabolites which are critically involved in immune responses. Female BALB/c mice are subjected to daily restraint stress to generate a model of depression and the growth of 4T1 mammary tumors is analyzed.

The authors convincingly show that stressed, tumor bearing mice have reduced levels of *Blautia* and its metabolite acetate. Further, treatment of stressed, tumor bearing mice with either strains of *Blautia* or acetate can negate the tumor promoting effects of chronic stress.

Additionally, these levels correlate with tumor infiltration by CD8⁺ T cells. Using patient material, they show that levels of acetate are significantly reduced in BreCa patients with depression compared to non-depressed patients.

Overall, this is a "proof of principal" paper showing that a specific strain of bacteria and one of its metabolites is reduced by stress and can be used therapeutically to overcome the tumor promoting effects of stress.

1. Many questions remain, including whether *Blautia* plays the same role in other strains of mice and tumor models. Also, it remains to be clarified how stress itself modulates the microbiome.

Response: We appreciate this kind suggestion which is very helpful for strengthening our findings by validating our key conclusions in additional models. We have established an additional depression-breast cancer model using different strain of mice and tumor cell to further validate our results. Specifically, we have developed a new depression-breast cancer model, wherein C57BL/6 mice were subjected to chronic unpredictable stress and injected with E0771 cells (Figure S1b). Our results from the newly established depression-breast cancer model are consistent with those obtained from our previous study, indicating that chronic stress exposure leads to faster tumor progression in comparison to non-stressed mice. In addition, stressed mice showed reduced numbers and functionality of tumor-infiltrating CD8⁺ T cells that produce inflammatory cytokines such as interferon gamma (INF- γ), as shown in Figure 1 of the revised manuscript. Furthermore, our findings reveal that the abundances of *Blautia coccoides* and *Blautia obeum* were significantly reduced in mice exposed to stress alone or stress in combination with tumor, resulting in a decrease in acetate content in tumor tissues (as illustrated in Figure S3d and Figure 5g in the revised manuscript).

Fig. S1. (b) Effects of chronic unpredictable stress (CUS) on body weight and immobility time of mice in the forced swim test (FST) and time in closed arms in elevated plus maze (EPM) test, respectively (n=15 per group).

Fig. 1. Chronic stress promotes breast cancer progression and impairs intratumoral CD8⁺ T cells. (a) Experimental design for Balb/c mice: one group was subjected to chronic restraint stress (RS) for 14 consecutive days before inoculation with 4T1 cells, and the other group was inoculated with 4T1 cells without stress exposure (n=10-11 per group). (b) Experimental design for C57 mice: one group was exposed to chronic unpredictable stress (CUS) for 14 consecutive days before inoculation with E0771 cells, and the other group was inoculated with E0771 cells without stress exposure (n=7-9 per group). (c) Tumor growth volume, weight, and lung metastasis burden in mice with 4T1 cells inoculation. (d) Tumor growth volume, weight, and lung metastasis burden in mice with E0771 cells inoculation. (e) Tumor-infiltrating CD8⁺ T cells in mice with 4T1 cells inoculation analyzed by flow cytometry (n=5 per group). (f) IFN- γ expression in tumor-infiltrating CD8⁺ T cells of mice with 4T1 cells inoculation analyzed by flow cytometry or qPCR (n=5 per group). (g) Tumor-infiltrating CD8⁺ T cells in mice with E0771 cells inoculation analyzed by flow cytometry (n=5 per group). (h) IFN- γ expression in tumor-infiltrating CD8⁺ T cells of mice with E0771 cells inoculation analyzed by flow cytometry (n=5 per group). Data are presented as mean \pm SEM. Statistical significance was determined using two-way ANOVA followed by Sidak's multiple comparison test for tumor volume, and unpaired two-tailed Student's t-test for the other data. Significance levels are denoted as * p < 0.05; ** p < 0.01; *** p < 0.001; **** p < 0.0001. "ns" indicates no significant difference.

Fig. S3. (d) Analysis of gene expression representative of *Blautia coccoides* and *Blautia obeum* in fecal samples from CUS mice by qPCR (n=6 per group).

Fig. 5. (g) Concentration of acetate in the tumor tissue of tumor and stressed (CUS) tumor mice (n=6 per group).

2. Abstract: The abstract states that comparison of BreCa patients indicated that depressed patients had reduced abundance of *Blautia* and acetate compared to non-depressed patients, but does not say whether the tumor sizes were larger or whether the prognosis or outcome is worse? If this information is available, please add it to the manuscript and at least, the Discussion.

Response: We agree with the reviewer's comment, together with similar comments from other reviewers, and have therefore increased the sample size for the breast cancer human cohort. We have also obtained the prognosis information of both depressed and non-depressed breast cancer patients. Our findings suggest that breast cancer patients with depression display lower levels of acetate, reduced populations of tumor-infiltrating CD8⁺ T cells, and an elevated risk of metastasis. These results are visually presented in Figure 7 of the revised manuscript. We have also incorporated these updated results into the abstract and discussion sections of the revised manuscript.

Fig. 7. Acetate levels correlate with CD8⁺ T cells in breast cancer patients. (a) Serum acetate levels in breast cancer patients with and without comorbid depression (n=28 and n=24, respectively). (b) Immunohistochemical staining of CD8⁺ T cells on breast cancer tissue samples obtained from

patients with and without comorbid depression (n=17 and n=11, respectively). (c) Spearman correlation analysis between serum acetate levels and the count of tumor-infiltrating CD8⁺ T cells (n=28). (d) The relationship among serum acetate levels, count of tumor-infiltrating CD8⁺ T cells, and the depression score of breast cancer patients. Each sample (n=28) was represented by a dot, with the dot's size indicating the CD8⁺ T cell count and its color representing the serum acetate levels. (e) Kaplan-Meier metastasis-free survival curves for breast cancer patients with and without comorbid depression (n=28 and n=24, respectively). Statistical significance was determined by unpaired two-tailed Student's t-test. Significance levels are denoted as * $p < 0.05$; ** $p < 0.01$; *** $p < 0.001$; **** $p < 0.0001$.

3. Use of the term “regression”. Regression implies that the tumors were large and became smaller with treatment. A different word would better convey the observation that the tumor-growth promoting effects of stress could be overcome by treating stressed, tumor bearing mice with *Blautia* or acetate. “Negates” is a more accurate and this wording should be changed in multiple places. The authors have used “eliminates” in some instances, and that is also a good choice.

Response: Thank you for this suggestion. In the revised manuscript, we have used "negate" or "eliminate" instead of "regress" to more accurately convey the intended meaning.

4. Throughout the manuscript, there are places where the text is written “*Blautia*-produced acetate underlies chronic stress promoted breast cancer progression”. This implies that there is a direct relationship and that *Blautia* is mediating the progression when the authors mean the opposite- that *Blautia* prevents stress induced progression. Please revise similar to how it is phrased in line 295: “Chronic stress leads to a gradual loss of *Blautia* and thereby reduced production of its metabolite acetate, which may underlie the reduced infiltration and impaired function of CD8⁺ T cells in the tumor, thereby promoting breast cancer growth and metastasis.”

Response: Thank you for this detailed suggestion. In the revised manuscript, we have corrected such sentences to avoid such confusion.

5. Co-housing experiments: Please explain the rationale for this and the experimental design more clearly. I presume the co-housing is to make sure the microbiomes are the same? Was this assessed?

Response: In our revised manuscript, we have provided a more detailed explanation for the rationale of the co-housing experiments. Our aim was to identify the causal microbiota for chronic stress-induced cancer progression. Co-housing approach has been widely applied to determine the causal effects of gut microbiome in certain phenotypes/pathophysiological events. It has been well-established that microbiota can be horizontally transferred from one mouse to another by co-housing the animals. In our study, we co-housed stressed tumor mice and control mice in gang cages at a 1:1 ratio until the end of the experiment. We believe that our co-housing approach allowed us to compare the microbial communities of the stressed and non-stressed mice in a controlled manner, thereby enabling us to identify the potential microbiota associated with chronic stress-induced cancer progression.

6. IL-6 and vegfc levels are reported, but there is not explanation of why and they aren't included in the conclusions. Please expand on the significance of these factors, or omit them.

Response: In the revised manuscript, we have revised the results description. Since previous studies have shown that IL-6 and vegfc are indicative of inflammation and metastasis respectively, they were determined in our study as a support for inflammation and metastasis of the models applied.

7. Line 127-128. Suggested re-wording...

“Results showed that the accelerated tumor growth-promoting effects of stress of breast cancer was regressed were lost when stressed mice were in co-housed with non-stressed mice, stress-treated tumor mice (Fig. 2h-2i), along with decreased expression levels of IL-6 and vegfc (Fig. S2d-2e)”.

Response: Thanks for this detailed suggestion. In the revised manuscript, we have rewritten this sentence to “Our results demonstrated that the accelerated progression of breast cancer induced by chronic stress was abolished when the stressed tumor mice were co-housed with non-stressed mice (Fig. 2f-2g and Fig. S2f-S2g)”.

8. Line 141 “microbe-phenotype triangulation”- Could this be briefly explained?

Response: Thanks for this suggestion. The “microbe-phenotype triangulation” approach has been recently developed by *Surana, N.K et al*^[1] and modified in our study. We have provided a more detailed description of our methodology in the revised manuscript. The microbe-phenotype triangulation approach had been recently developed to pinpoint microbes that are more likely to be causally related to disease. This approach involves integrating and analyzing various data sources, such as microbiome and host phenotype data, to establish causal relationships between specific microbes and host phenotype. By reducing noise in microbiota studies, this approach is an effective means of identifying the causal microbes involved in diseases. In light of this, we extended to develop a spatiotemporal microbe-phenotype triangulation approach, which combines metagenomic and metabolomic analysis at vertical spatial sequence with different time points at horizontal temporal series for precising identification of causal microbes and metabolites.

Reference:

[1] Surana, N.K. & Kasper, D.L. Moving beyond microbiome-wide associations to causal microbe identification. *Nature* 552, 244-247 (2017).

9. Figures, the font is too small and extremely hard to read (6pt in figures as printed).

Response: OK. We have revised all the figures according to your suggestions in the revised manuscript.

10. Line 201: In addition, the supplemented strains of *Blautia* (*Blautia coccoides* and *Blautia obeum*) that significantly regressed negated cancer progression by chronic stress promoted cancer progression were also able to produce large amounts of acetate (Fig. S4f). There are several places where *Blautia* is misspelled including line 28 in the abstract.

Response: Thanks. We have made the necessary corrections as suggested in the revised manuscript.

11. “Acetate supplementation overcomes chronic stress-promoted breast cancer progression by activating CD8⁺ T cells.

Response: We have implemented the recommended corrections in the revised manuscript according to your suggestions.

12. . Fig 7- define hpf

Response: We have detailed HPF in the revised manuscript. CD8-positive cells within tumors were quantified by two independent pathologists through manual counting in five high-powered fields (HPFs).

13. Fig 8- Title should read Deficiency in *Blautia*-produced acetate underlies chronic stress promoted breast cancer progression. In present study, using an effective approach which combined temporospatial mapping of gut microbiota and its metabolites with metabolomics and metagenomics, we revealed that chronic stress results in reduced numbers of the causal microbe (*Blautia*) and its metabolite (acetate). These deficiencies may underlie chronic stress promoted breast cancer progression by regulating reducing number and function of CD8⁺ T cells function in the tumor microenvironment.

Response: We have thoroughly reviewed the manuscript and have made the revisions based on your suggestions.

Reviewer #2 (Remarks to the Author): with expertise in breast cancer, microbiota

In the manuscript of ‘Repressed *Blautia*-acetate immunological axis underlies chronic stress promoted breast cancer progression’, Ye et al. addressed an interesting question of how chronic stress impinges on the gut microbiome and whether this stress associated gut microbiome IFN- γ promote tumor growth and lung metastasis, and this tumor promoting effect was dependent on the gut microbiome, as depletion of gut microbiome abolished the stress mediated effect. After analysis of the gut microbiome composition, they pinpointed that *Blautia* was the key bacteria in the gut microbiota mediating stress associated tumor growth and metastasis. Mechanistically, they reported that *Blautia*’s metabolite acetate can promote CD8⁺ T cells infiltration and activation, therefore reduced *Blautia*-acetate axis accelerate the progression of breast cancer by chronic stress. In addition, they found acetate was correlated with CD8⁺ T cells in breast cancer patients. The scientific question they asked is an interesting one and is important for the cancer field. However, I am not fully convinced due to some experimental flaws and some inconsistencies.

My major criticisms are listed below:

1. Figure 1g, 2K, Please show the gating strategy. Contour plot is not good to show the positive population, please show the dot plot, also show the negative controls. From the contour plot, the IFN γ staining does not seem to work.

Response: Thank you for your instructive suggestions. We have included the gating strategy for the detection of IFN- γ by flow cytometry in Figure S6 and presented the positive population of IFN- γ staining using a dot plot in the revised manuscript. As shown in Figure 1f, the positive population of IFN- γ staining was significantly lower in the stressed tumor group compared to the non-stressed tumor group, and the relative mRNA expression of IFN- γ was also significantly decreased in the stressed tumor group. We also observed similar results in our other depression-breast cancer model using C57BL/6 mice exposed to chronic unpredictable stressors and injected with E0771 breast cancer cells (Figure 1h). These results suggest compromised responses of infiltrated CD8⁺ T cells in the presence of comorbid tumor and depression.

Fig. S6. Gating strategy for the detection of IFN- γ by flow cytometry.

Fig. 1. (e) Tumor-infiltrating CD8⁺ T cells in mice with 4T1 cells inoculation analyzed by flow cytometry (n=5 per group). (f) IFN- γ expression in tumor-infiltrating CD8⁺ T cells of mice with 4T1 cells inoculation analyzed by flow cytometry or qPCR (n=5 per group). (g) Tumor-infiltrating CD8⁺ T cells in mice with E0771 cells inoculation analyzed by flow cytometry (n=5 per group). (h) IFN- γ expression in tumor-infiltrating CD8⁺ T cells of mice with E0771 cells inoculation analyzed by flow cytometry (n=5 per group).

2. *Blautia* might not be the only acetate producing bacteria in the gut, why other acetate producing bacteria can not compensate the effect of *Blautia* reduction?

Response: Thanks for this instructive comment. We agree with this comment that there are other potential acetate-producing bacteria in addition to *Blautia* might be also applied to compensate for *Blautia* reduction. To address this concern, we have added some discussions: Although there are many enteric bacteria that produce acetate^[1-2], *Blautia* is one of the most abundant genera in the gut, regardless of race^[3-5], and is a key member of the Lachnospiraceae family that accounts for 50% of the total intestinal microbiota^[6]. Moreover, a previous study examining the acetate-producing capabilities of 110 Lachnospiraceae strains found that 91 strains produced acetate, and the top five acetate producers were all from *Blautia*^[7-8]. Our study revealed that *Blautia* was significantly reduced in conditions of depression, indicating that it is the most important acetate producer in the gut. In light of this evidence, we propose that the supplementation with *Blautia* is an effective approach to compensate for the reduction of acetate production in the gut.

References:

[1] Koh A, De Vadder F, Kovatcheva-Datchary P, Bäckhed F. From Dietary Fiber to Host Physiology: Short-Chain Fatty Acids as Key Bacterial Metabolites. *Cell*. 2016 Jun 2;165(6):1332-1345. doi: 10.1016/j.cell.2016.05.041. PMID: 27259147.

- [2] Petra Louis, Georgina L Hold, Harry J Flint. The gut microbiota, bacterial metabolites and colorectal cancer. *Nat Rev Microbiol.* 2014 Oct;12(10):661-72.
- [3] Nishijima, S. et al. The gut microbiome of healthy Japanese and its microbial and functional uniqueness. *DNA Res.* 23, 125–133 (2016).
- [4] Bamberger, C. et al. A walnut-enriched diet affects gut microbiome in healthy Caucasian subjects: a randomized, controlled trial. *Nutrients* 10, 244 (2018).
- [5] Zhang, W. et al. Gut microbiota community characteristics and disease-related microorganism pattern in a population of healthy Chinese people. *Sci. Rep.* 9, 1594 (2019).
- [6] Biddle A, Stewart L, Blanchard J, Leschine S. Untangling the genetic basis of fibrolytic specialization by lachnospiraceae and ruminococcaceae in diverse gut communities. *Diversity.* 2013;5(3):627–640. doi:10.3390/d5030627.
- [7] Abdugheni, R., Wang, W.-Z., Wang, Y.-J., Du, M.-X., Liu, F.-L., Zhou, N., Jiang, C.-Y., Wang, C.-Y., Wu, L., Ma, J., Liu, C., Liu, S.-J. Metabolite profiling of human-originated Lachnospiraceae at the strain level. *iMeta.* 2022 Dec; 1(4): e58.
- [8] Vacca M, Celano G, Calabrese FM, Portincasa P, Gobbetti M, De Angelis M. The Controversial Role of Human Gut Lachnospiraceae. *Microorganisms.* 2020 Apr 15;8(4):573. doi: 10.3390/microorganisms8040573.

3. The effect of ABX treatment on gut microbiome was not quantified. It is hard to judge whether it is complete abolishment of gut microbiome or it is just dysbiosis.

Response: We have now quantified the impact of the antibiotic (ABX) treatment on the gut microbiome and observed a complete elimination of microbial populations (Figure S2a-S2b).

Figure S2. (a) Antibiotic-induced microbiome depletion illustrated through sample stool cultures of control and pseudo mice. (b) DNA extraction performed on stool samples from both control mice and pseudo mice (n=4 per group).

4. Figure 3a-c and 3d-f are both co-housing experiments, however, the bacteria dynamics are quite different, *Ruminiclostridium* showed significant change in one experiment but not the other. Lachnospiraceae and ruminococcus showed up in one but not the other. How variable is this experiment?

Response: Thanks for this reminding comment. The data shown in Figure 3a-3c and Figure 3d-3f were obtained from distinct experimental protocol. To identify exact causal microbiota and metabolites, we innovated an temporospatial microbe-phenotype triangulation approach. We initially screened for microbes that exhibited significant changes upon co-housing along the spatial axis, which involved analyzing cecum samples from different groups (Figure 3a-3c). We identified several microbes that were significantly altered, including *Blautia*, *Ruminiclostridium*, and *Intestinimonas*. To verify our list of potential causal microbes, we further investigated the

abundance of microbes over time. We reasoned that only the microbiota exhibiting stable and time-dependent changes were truly causal. Therefore, we analyzed fecal samples from the co-housed stressed tumor group at different time points along the temporal axis. We observed a gradual increase in the abundance of *Blautia*, *Ruminiclostridium*, *Ruminococcus*, and *Lachnospiraceae_NK4A136_group* from day 14, with maximal changes seen after 40 days (Figure 3d-3f). This temporospatial approach allowed us to identify *Blautia* and *Ruminiclostridium* as potential causal microbes responsible for facilitating depression-related tumor progression. We suppose that the reviewer may agree, that the gut microbiota are highly variable and dynamic, and therefore it is understandable that different results might be obtained from different batch of experiments. In view of this challenge, the temporospatial microbe-phenotype triangulation approach performed in our study could be expected to be reliable to ascertain the exact causal microbiota underlying various pathophysiological events.

5. In Figure S3b, tumor graft seems to increase the *Blautia* abundance, which is supposed to enhance T cell activity, would it be the opposite? In addition, the *Blautia* level in Tumor⁺stress group is reduced to a similar level to wt control group, why co-housing of the two mice with similar *Blautia* level can rescue the Tumor-stress *Blautia* level?

Response: Thanks for this concern. In our analysis of 16S ribosomal RNA (rRNA) gene sequencing, we compared the abundance of *Blautia* in the control and tumor groups and found no significant difference in *Blautia* abundance between the control and tumor groups (Fig.S3b). It is important to note that the *Blautia* level was determined at the end of 40 days with and without cancer, and thus these data should not be directly comparable to those obtained from the co-housing experiments. To address this concern, we further analyzed the data on *Blautia* abundance at the species level and found that the level of *Blautia spp.* in the stressed tumor group (Average=0.0006014) showed a further decreased trend compared with the control group (Average=0.0018654) (Fig. S3b). Moreover, combined with the changes observed in our co-housing experiment, we observed that *Blautia spp.* may be related to chronic stress-promoted breast cancer progression (Fig. S3c). We also confirmed the relevance of *Blautia spp.* to chronic stress through qPCR analysis, which showed that the abundances of *Blautia coccoides* and *Blautia obeum* in the control group were significantly higher than those in the stressed tumor group in the CUS model (Fig. S3d). These additional data have been included in the revised manuscript.

Fig. S3. (b) Relative abundance of *Blautia* at the genus level in the cecal sample by 16S rRNA sequencing. (c) Relative abundance of *Blautia spp.* in the cecal sample by 16S rRNA sequencing.

(d) Analysis of gene expression representative of *Blautia coccoides* and *Blautia obeum* in fecal samples from CUS mice by qPCR.

6. From Figure 7a, the concentration of acetate in breast cancer patients' serum is 25-160ug/ml, and mouse serum and tumor tissue showed similar concentration, while in Figure 6, NaAc was given 1g/kg intraperitoneally per day, which is way more than the physiological concentration. In addition, Figure 5f showed acetate level in tumor tissue, stress only reduced acetate level by ~20% with big variation, and 20% of acetate alteration can cause striking phenotype, why need to administer such a high level of NaAc to rescue the phenotype?

Response: Thanks for this concern. Acetate can be supplemented to animals through various routes, including drinking water, diet, and intraperitoneal injection^[1-3]. We choose a dose of 1 g per kg body weight of sodium acetate by intraperitoneal injection based on previous reports^[3,4]. It has been shown that intraperitoneal injection of sodium acetate at a dose of 1g/kg can lead to increased serum acetate levels in mice, with a range of variation from 1-70% and an average increase of approximately 45%^[4]. This increase in serum acetate levels is comparable to the decrease in acetate levels induced by chronic stress, which was found to be an average of approximately 20% in our study. According to these previous evidences, we suppose that the administration of 1g/kg acetate sodium to the mice is pathophysiologically relevant. To address this concern, we have added some discussions by citing appropriate references in the revised manuscript.

References:

- [1] Daniel Erny., et al. Microbiota-derived acetate enables the metabolic fitness of the brain innate immune system during health and disease. *Cell Metab.* 2021 Nov 2;33(11):2260-2276.e7.
- [2] Ying Hong., et al. *Desulfovibrio vulgaris*, a potent acetic acid-producing bacterium, attenuates nonalcoholic fatty liver disease in mice. *Gut Microbes.* 2021 Jan-Dec;13(1):1-20.
- [3] Trompette A., et al. Gut microbiota metabolism of dietary fiber influences allergic airway disease and hematopoiesis. *Nat Med.* 2014;20(2):159–66.
- [4] Mateus B Casaro., et al. A probiotic has differential effects on allergic airway inflammation in A/J and C57BL/6 mice and is correlated with the gut microbiome. *Microbiome.* 2021 Jun 10;9(1):134.

7. According to the model, *Blautia*- acetate-T cell axis mediated chronic stress associated breast cancer progression. Therefore, *Blautia* and acetate should be the downstream of stress. However, ABX treatment, *Blautia* inoculation and acetate supplement were performed before chronic stress exposure and tumor inoculation. Do these treatments influence tumor initiation or tumor progression?

Response: Thanks for this concern. We agree with the reviewer's comment that *Blautia* and acetate should be the downstream of stress. To validate the causal effects of *Blautia* and acetate in chronic stress aggravated tumor initiation/progression, *Blautia* and/or acetate were directly administered to mice with tumor xenograft. In our study, ABX treatment, *Blautia* inoculation and acetate supplement were applied to phenocopy and/or combat chronic stress to tumor progression. Such experimental protocols had been widely designed to validate potential causal effects of microbiota and the associated metabolites were relevant to certain pathophysiological events. Results obtained from these experiments support that chronic stress aggravated tumor progression is gut microbiota dependent and *Blautia*-acetate is possibly a causal factor. By performing these treatments before chronic stress exposure and tumor inoculation, the study mainly aimed to establish a causal

relationship between the axis and depression-promoted tumor progression, rather than to directly assess the impact of the treatments on tumor initiation or progression. Nonetheless, we agree with the reviewer's concern, the potential impact of these treatments on tumor development should be taken into consideration when interpreting the results. We have now revised the description of the results in the revised manuscript.

8. Figure 7b. The high-resolution pictures should be provided. IHC staining of CD8⁺ T cell in depressed breast cancer patients' tissues seems the adjacent normal tissue. and the breast cancer types should be recorded in a table.

Response: We have provided high-resolution pictures in the revised manuscript and the breast cancer types of the breast cancer patients were invasive ductal carcinoma or invasive lobular carcinoma.

Fig. 7. Acetate levels correlate with CD8⁺ T cells in breast cancer patients. (a) Serum acetate levels in breast cancer patients with and without comorbid depression (n=28 and n=24, respectively). (b) Immunohistochemical staining of CD8⁺ T cells on breast cancer tissue samples obtained from

patients with and without comorbid depression (n=17 and n=11, respectively). (c) Spearman correlation analysis between serum acetate levels and the count of tumor-infiltrating CD8⁺ T cells (n=28). (d) The relationship among serum acetate levels, count of tumor-infiltrating CD8⁺ T cells, and the depression score of breast cancer patients. Each sample (n=28) was represented by a dot, with the dot's size indicating the CD8⁺ T cell count and its color representing the serum acetate levels. (e) Kaplan-Meier metastasis-free survival curves for breast cancer patients with and without comorbid depression (n=28 and n=24, respectively). Statistical significance was determined by unpaired two-tailed Student's t-test. Significance levels are denoted as * $p < 0.05$; ** $p < 0.01$; *** $p < 0.001$; **** $p < 0.0001$.

Minor concerns are:

1. There are some inconsistencies in tumor growth: Figure 1b, Figure 4c and Figure 6b, the tumor started to growth from day 10 while in Figure 2b, it was around day 7. Moreover, at day 26, why the tumor volume of tumor, tumor⁺stress group are different in Figure 1b, Figure 2b, Figure 4c and Figure 6b. Is the stress model stable?

Response: The stress-tumor model employed in our study had been previously described in several studies [1-3]. Following tumor cell inoculation, we monitored tumor growth every other day. As noted by the reviewer, some variations in the initiating time point and the volume of tumor were observed from different batches of animal studies. Such variations can be found in many other cancer studies [4-5]. We suppose such experimental variations in different batches of studies are reasonable. Since appropriate controls have been designed and most of the experiments have been reproduced at least twice, it seems that such variations observed from different batches of studies may not compromise the conclusions. Moreover, we have now added another stress-tumor model to further validate our findings, and reliable and reproducible results have been obtained from these experiments.

References:

- [1] Thaker, P.H., et al. Chronic stress promotes tumor growth and angiogenesis in a mouse model of ovarian carcinoma. *Nat Med* 12, 939-944 (2006).
- [2] Chronic stress in mice remodels lymph vasculature to promote tumour cell dissemination. *Nat Commun* 7, 10634 (2016).
- [3] Kim-Fuchs, C., et al. Chronic stress accelerates pancreatic cancer growth and invasion: A critical role for beta-adrenergic signaling in the pancreatic microenvironment. *Brain Behavior & Immunity* 40, 40-47 (2014).
- [4] He, Y., et al. Gut microbial metabolites facilitate anticancer therapy efficacy by modulating cytotoxic CD8(+) T cell immunity. *Cell Metab* 33, 988-1000 e1007 (2021).
- [5] Yan Li., et al. Gut microbiota dependent anti-tumor immunity restricts melanoma growth in *Rnf5*^{-/-} mice. *Nat Commun*. 2019 Apr 2;10(1):1492.

2. There are some typos, line 28, 173, 177, 201, 205, 742 “Blatua”, should be “*Blautia*”; line 787 “corrected” should be “correlated”.

Response: Thank you for pointing out our mistakes. We have corrected them in the revised manuscript.

3. The phylum and genus distribution of gut microbiota in each condition for each individual mouse should be shown in the supplementary figures for better judgement of the data quality.

Response: We have included the phylum and genus distribution of gut microbiota in each condition for each individual mouse in the supplementary figures (figure S3e-S3h).

Fig. S3. (e-f) Average relative abundance of representative phylum (e) and genus (f) in the gut microbiota from the co-housing experiment by 16S rRNA sequencing. (g-h) Time-dependent changes in relative abundance at the phylum (g) and genus (h) level from the fecal microbiota of co-housed mice by 16S rRNA sequencing.

4. The color label in Figure 3b for Peoptococcus is not consistent with Figure 3C bar plot.

Response: We have corrected it in the revised manuscript.

Reviewer #3 (Remarks to the Author): with expertise in metagenomics, metabolomics

Remarks to the Author:

In this study, titled "Repressed *Blautia*-acetate immunological axis underlies chronic stress promoted breast cancer progression" by Ye et al, the authors used a navigational strategy called "microbe-phenotype triangulation" to pinpoint casual protective microbe that shape host physiology and disease susceptibility. They identified *Blautia* and its metabolite acetate as protective factors against breast cancer progression under chronic stress. The underlying mechanism is to promote anti-tumor immune response by restoring the function of tumor infiltrating CD8⁺ T cells. The study is well described and structured. Find below major comments which, if addressed, would significantly strengthen the manuscript.

Introduction

1. It is recommended to first present global epidemiological data on breast cancer, including incidence, mortality, etc. Based on previous studies, the authors should then list the pathogenesis of breast cancer and then describe the impact of depression as one of the factors on the development of breast cancer. Finally, the *Blautia*-acetate immunological axis was introduced, focusing on its role in possible influence in the development of breast cancer.

Response: Thank you for your valuable suggestions. In the introduction section of our revised manuscript, we initially present the global epidemiological data on breast cancer, followed by highlighting the impact of depression on breast cancer development. We then discuss the current status and research gaps concerning depression promoted cancer progression, and finally, we elucidate the potential role of the intestinal microbiota-metabolite immunological axis in depression promoted cancer progression. It's worth noting that while *Blautia*-acetate is a causal microflora and metabolite discovered in our research, we did not introduce it in the introduction section. Instead, we focused on discussing it in the later discussion session.

Methods

1. As far as I know the common mouse models of depression include stress modeling, drug modeling, surgical modeling and genetic modeling. Chronic restraint stress mice can partially describe a state of psychological stress but are prone to adaptation. Whether the use of other experimental animal models could better validate the phenomenon found by the authors? I have seen only one breast cancer murine cell line used by the authors, and I think at least two different cell lines are needed for validation.

Response: We appreciate this kind suggestion which is very helpful for strengthening our findings by validating our key conclusions in additional models. We have established an additional depression-breast cancer model using different strain of mice and tumor cell to further validate our results. Specifically, we have developed a new depression-breast cancer model, wherein C57BL/6 mice were subjected to chronic unpredictable stress and injected with E0771 cells (Figure S1b).

Our results from the newly established depression-breast cancer model are consistent with those obtained from our previous study, indicating that chronic stress exposure leads to faster tumor progression in comparison to non-stressed mice. In addition, stressed mice showed reduced numbers and functionality of tumor-infiltrating CD8⁺ T cells that produce inflammatory cytokines such as interferon gamma (IFN- γ), as shown in Figure 1 of the revised manuscript. Furthermore, our

findings reveal that the abundances of *Blautia coccoides* and *Blautia obeum* were significantly reduced in mice exposed to stress alone or stress in combination with tumor, resulting in a decrease in acetate content in tumor tissues (as illustrated in Figure S3d and Figure 5g in the revised manuscript).

Fig. S1. (b) Effects of chronic unpredictable stress (CUS) on body weight and immobility time of mice in the forced swim test (FST) and time in closed arms in elevated plus maze (EPM) test, respectively (n=15 per group).

Fig. 1. Chronic stress promotes breast cancer progression and impairs intratumoral CD8⁺ T cells. (a) Experimental design for Balb/c mice: one group was subjected to chronic restraint stress (RS) for 14 consecutive days before inoculation with 4T1 cells, and the other group was inoculated with 4T1 cells without stress exposure (n=10-11 per group). (b) Experimental design for C57 mice: one group was exposed to chronic unpredictable stress (CUS) for 14 consecutive days before inoculation with E0771 cells, and the other group was inoculated with E0771 cells without stress exposure (n=7-9 per group). (c) Tumor growth volume, weight, and lung metastasis burden in mice with 4T1 cells inoculation. (d) Tumor growth volume, weight, and lung metastasis burden in mice with E0771 cells inoculation. (e) Tumor-infiltrating CD8⁺ T cells in mice with 4T1 cells inoculation analyzed by flow cytometry (n=5 per group). (f) IFN- γ expression in tumor-infiltrating CD8⁺ T cells of mice with 4T1 cells inoculation analyzed by flow cytometry or qPCR (n=5 per group). (g) Tumor-infiltrating CD8⁺ T cells in mice with E0771 cells inoculation analyzed by flow cytometry (n=5 per group). (h) IFN- γ expression in tumor-infiltrating CD8⁺ T cells of mice with E0771 cells inoculation analyzed by flow cytometry (n=5 per group). Data are presented as mean \pm SEM. Statistical significance was determined using two-way ANOVA followed by Sidak's multiple comparison test for tumor volume, and unpaired two-tailed Student's t-test for the other data. Significance levels are denoted as * p < 0.05; ** p < 0.01; *** p < 0.001; **** p < 0.0001. "ns" indicates no significant difference.

Fig. S3. (d) Analysis of gene expression representative of *Blautia coccoides* and *Blautia obeum* in fecal samples from CUS mice by qPCR (n=6 per group).

Fig. 5.(g) Concentration of acetate in the tumor tissue of tumor and stressed (CUS) tumor mice (n=6 per group).

Results

1. The description of the results needs to be concise and no specific analysis is needed here, e.g., the body weight of mice in group A is higher than that in group B, and the interferon γ of mice in group A is lower than that in group B. Detailed description of the treatment of mice in the method section, and only a brief indication in the results is needed.

Response: We have revised the results based on your professional suggestions in the revised manuscript.

2. In Figure 5e&f the authors showed acetate and stearylcarntine in serum and tumor samples represented the most significantly changed metabolites by co-housing. The authors are requested to specifically label these two metabolites in Fig 5c and d to allow us to better see the variation in their levels in different groups.

Response: In the revised version of the manuscript, we have added explicit labels to clearly indicate these elements.

3. Is acetate produced exclusively by *Blautia*, or do other bacteria also metabolize acetate? How important is the role of *Blautia* in this, and is it possible to knock out the enzyme in which *Blautia* metabolizes acetate, or is it possible to specifically delete the bacteria?

Response: (1) Thanks for this instructive comment. We agree with this comment that there are other potential acetate-producing bacteria in addition to *Blautia* might be also applied to compensate for *Blautia* reduction. To address this concern, we have added some discussions: Although there are many enteric bacteria that produce acetate^[1-2], *Blautia* is one of the most abundant genera in the gut, regardless of race^[3-5], and is a key member of the Lachnospiraceae family that accounts for 50% of the total intestinal microbiota^[6]. Moreover, a previous study examining the acetate-producing capabilities of 110 Lachnospiraceae strains found that 91 strains produced acetate, and the top five acetate producers were all from *Blautia*^[7-8]. Our study revealed that *Blautia* was significantly reduced in conditions of depression, indicating that it is the most important acetate producer in the gut. In light of this evidence, we propose that the supplementation with *Blautia* is an effective approach to compensate for the reduction of acetate production in the gut.

(2) *Blautia* is a well-known acetogen, and our study demonstrated that both *Blautia* and acetate supplementation can reverse stress-promoted breast cancer progression. As you mentioned, generating a mutant strain of *Blautia* that cannot produce acetate would be crucial for analyzing the in vivo effects of *Blautia*-acetate. However, one of the main challenges in modifying *Blautia*'s metabolism is the lack of effective genetic modification tools due to its thick cytoderm, which makes it difficult for current gene-editing techniques to penetrate and mutate target genes involved in acetate production. Additionally, we were unsuccessful in finding an antibiotic to selectively deplete *Blautia*.

(3) To address this concern, we used shotgun metagenomics to determine the alteration of the enzymes of *Blautia* involved in acetate metabolism. Our metagenomic data indicated that the Wood-Ljungdahl pathway, which is the main pathway generating acetate by *Blautia*^[9], was significantly downregulated, suggesting decreased acetyl-CoA capacity (Fig. 5i-5j). We observed that nearly all

the key genes involved in CO₂ fixation (K00198, K15022), formyl-tetrahydrofolate ligase (K01491, K00297), and acetyl-CoA synthase (K01438) were significantly reduced (Fig. 5k), supporting the reduction of acetyl-CoA biosynthesis upon chronic stress. Further analysis revealed striking changes in pyruvate metabolism including phosphoacetylase (K00625) and acetate kinase (K00925) in stressed tumor mice (Fig. 5k). To identify the bacterial genera involved in Wood-Ljungdahl pathway, we conducted an analysis of species contributing to K00198 from the top 50 bacterial genera (Fig. S4f), and it was determined that the crucial K00198 was present in *Blautia*, *Lachnoclostridium*, *Ruminococcus*, *Clostridium*, and *Oscillibacter* genera. Taken together, these findings support that, under conditions of chronic stress, the decreased abundance of *Blautia* is more likely in contribution to the reduction of acetate production, and that the disruption of *Blautia*-acetate axis is likely involved in the pathological progression of breast cancer promoted by chronic stress.

Fig. 5. (i) Linear discriminative analysis (LDA) Effect Size (LEfSe) analysis between tumor group and stressed tumor group in KEGG modules. (j) Overview of the Wood-Ljungdahl pathway: K00198: anaerobic carbon-monoxide dehydrogenase catalytic subunit; K15022: formate dehydrogenase (NADP⁺) beta subunit; K01491: methylenetetrahydrofolate dehydrogenase (NADP⁺) / methylenetetrahydrofolate cyclohydrolase; K00297: methylenetetrahydrofolate reductase (NADH); K14138: acetyl-CoA synthase; K00625: phosphate acetyltransferase; K00925: acetate kinase. (k) The relative abundance KOs associated with Wood-Ljungdahl pathway (n=5 per group).

Fig. S4. (f) Barplot of species and functional contribution analysis, highlighting the top 50 genus contributing the k00198.

References:

[1] Koh A, De Vadder F, Kovatcheva-Datchary P, Bäckhed F. From Dietary Fiber to Host Physiology: Short-Chain Fatty Acids as Key Bacterial Metabolites. *Cell*. 2016 Jun 2;165(6):1332-1345. doi: 10.1016/j.cell.2016.05.041. PMID: 27259147.

[2] Petra Louis, Georgina L Hold, Harry J Flint. The gut microbiota, bacterial metabolites and colorectal cancer. *Nat Rev Microbiol*. 2014 Oct;12(10):661-72.

[3] Nishijima, S. et al. The gut microbiome of healthy Japanese and its microbial and functional uniqueness. *DNA Res*. 23, 125–133 (2016).

[4] Bamberger, C. et al. A walnut-enriched diet affects gut microbiome in healthy Caucasian subjects: a randomized, controlled trial. *Nutrients* 10, 244 (2018).

[5] Zhang, W. et al. Gut microbiota community characteristics and disease-related microorganism pattern in a population of healthy Chinese people. *Sci. Rep.* 9, 1594 (2019).

[6] Biddle A, Stewart L, Blanchard J, Leschine S. Untangling the genetic basis of fibrolytic specialization by lachnospiraceae and ruminococcaceae in diverse gut communities. *Diversity*. 2013;5(3):627–640. doi:10.3390/d5030627.

[7] Abdugheni, R., Wang, W.-Z., Wang, Y.-J., Du, M.-X., Liu, F.-L., Zhou, N., Jiang, C.-Y., Wang, C.-Y., Wu, L., Ma, J., Liu, C., Liu, S.-J. Metabolite profiling of human-originated Lachnospiraceae at the strain level. *iMeta*. 2022 Dec; 1(4): e58.

[8] Vacca M, Celano G, Calabrese FM, Portincasa P, Gobbetti M, De Angelis M. The Controversial Role of Human Gut Lachnospiraceae. *Microorganisms*. 2020 Apr 15;8(4):573. doi: 10.3390/microorganisms8040573.

[9] Schiel-Bengelsdorf, B. & Dürre, P. Pathway engineering and synthetic biology using acetogens. *FEBS letters* 586, 2191-2198 (2012).

4. The authors' findings suggested that acetate may have no direct effect on cancer cells and CD8⁺ T cells, but may prevent cancer-induced impairment of CD8⁺ T cells, thereby enhancing anti-tumor immunity. This is a very interesting phenomenon, can the author explain the mechanism or the possible reasons for it?

Response: Thanks for this comment.

our results revealed that acetate has no direct effects on CD8⁺ T cells but can restore the function of impaired CD8⁺ T cells when co-cultured with tumor cells. Although acetate may directly induce apoptosis in colorectal carcinoma cells at a high concentration of up to 70 mmol/L^[1-2], such levels are unlikely to be reached under real pathophysiological conditions. At pathophysiologically relevant concentration, acetate may have little direct toxic effects against tumor cells. Thus, our study together with previous findings indicate that acetate may take effects via interfering the interactions between cancer and T cells, rather than direct effects against either cancer or T cells. Moreover, RNA sequencing results obtained from CD8⁺ T cells co-cultured with 4T1 cells, revealed a limited number of differentially expressed genes by acetate treatment. Isotopic tracing study showed that CD8⁺ T cells can take up acetate and represent an important resource to feed the acetyl-CoA pool and thereby augmenting IFN- γ production, potentially enhancing the anti-tumor response (Fig.6j-6n). The exact molecular mechanisms underlying how acetate boosts anti-tumor T cell immunity in tumor environment warrants comprehensive research. The present results together with previous findings suggest that it is important to investigate how cancer cells and T cells compete for the direct usage of acetate, and how acetate functions as a signaling molecule in interfering and orchestrating interactions between cancer cells and T cells.

Fig. 6. (j) Heatmap (left panel) and MA-plot (right panel) of control and acetate-exposed CD8⁺ T cells under co-culture with 4T1 cells transcriptome (n=5 per group). (k) Pathway of acetate uptake and metabolism in CD8⁺ T cells. (l) mRNA expression of the solute carrier receptors *Slc16a1*, *Slc16a3*, *Acss1*, *Acss2*, and *Acly* in control and acetate-exposed CD8⁺ T cells under co-culture with 4T1 cells as determined by qPCR(n=4 per group). (m) Metabolic tracing analysis of CD8⁺ T cells under co-culture with 4T1 cells after exposure to ¹³C-acetate. The x-axis shows the number of ¹³C per respective metabolite. Depicted are pooled data from two independent experiments with cells from n=3 mice each. (n) Acetyl-CoA concentration in CD8⁺ T cell lysates after co-culture with 4T1 cells with or without acetate (n=5 per group).

References:

[1] Marques, C., et al. Acetate-induced apoptosis in colorectal carcinoma cells involves lysosomal membrane permeabilization and cathepsin D release. *Cell Death Dis* 4, e507 (2013).

[2] Zeng, H., Hamlin, S.K., Safratowich, B.D., Cheng, W.H. & Johnson, L.K. Superior inhibitory efficacy of butyrate over propionate and acetate against human colon cancer cell proliferation via cell cycle arrest and apoptosis: linking dietary fiber to cancer prevention. *Nutr Res* 83, 63-72 (2020).

5. I think it is a great pity that stool samples from patients with breast cancer combined with depression were not collected for validation in this study.

Response: Thanks for this comment. We agree with the reviewer that it is a limitation of our study failed to include stool samples from patients with breast cancer combined with depression. To address this concern, we have now added appropriate discussions: “One limitation of our study is the unavailability of fecal samples from breast cancer patients with and without depression, which hindered a direct assessment of how the *Blautia*-acetate axis is involved in these patients. However, results collected from animal studies, the in vitro co-culture system, and correlation analysis in human patients suggest that the compromised *Blautia*-acetate axis likely contributes to depression-promoted breast cancer progression in humans, and indicate that targeting the *Blautia*-acetate axis could be a promising therapeutic approach for breast cancer patients with co-existent depression.”

Discussion

1. The article focuses on the effect of *Blautia*-acetate immunological axis in breast cancer, which requires a detailed analysis of its underlying mechanism. Firstly, analyze the effect of *Blautia*-acetate immunological axis in other diseases, then analyze the effect of *Blautia*-acetate immunological axis in the context of the results of this study, confirm where your results agree or disagree with others, and analyze the reasons for the disagreement.

Response: Based on your professional suggestions, we have revised the discussion in the revised manuscript.

Reviewer #4 (Remarks to the Author): with expertise in cancer immunology

In this manuscript, Ye L. and colleagues postulated that the *Blautia*-generated SCFA acetate suppresses stress-promoted breast cancer progression by activating T cell-mediated anti-tumor immunity. The authors show increased frequency of tumor-infiltrated IFN-g-expressing CD8⁺ T cells following treatment of mice with acetate. Moreover, the acetate levels are positively associated with tumor-infiltrated T cells in breast cancer patients. While some of the results are potentially very interesting, there are several issues that should be thoroughly addressed.

Major concern:

1. The essential part of the paper is the acetate-induced tumor regression of stress-promoted tumor progression. It is important to analyze the effects of acetate on the survival and cytotoxic capacity of CTLs in the co-culture with 4T1 cancer cells in more detail. In contrast to butyrate, acetate appears not to be able to directly affect the functionality of CTLs. Further functional experiments are required to understand mechanisms underlying acetate-mediated effects. Metabolic flux analysis of co-cultured CTLs and 4T1 cells using ¹³C-labeled sodium acetate is needed to understand metabolic profile of cells following acetate treatment (high glucose vs. low glucose). The authors also should explore the relation between acetate treatment of co-cultures (as compared to CTLs without 4T1 cells) and possible alterations in glycolysis and OXPHOS. Are glucose-restricted CTLs in the co-culture more or less responsive to acetate-mediated effects as compared to normal CTLs? Some papers have shown a direct effect of acetate on CTLs (Luu Maik et al., *Sci. Rep.*, 2018), and acetate is even incorporated into histones of CTLs (Qiu Jing et al., *Cell Reports*, 2019). The authors should critically discuss the established literature (e.g. the paper: Systemic short chain fatty acids limit antitumor effect of CTLA-4 blockade in hosts with cancer, *Nature Comms.*, 2020), and they might also provide the data for other SCFAs in the co-cultures (CTLs and 4T1 cancer cells).

Response: Thanks for all of these instructive comments. As you have mentioned, panels of previous reports have shown that acetate may regulate the function of CD8⁺ T cells via diverse mechanisms, such as enhancing glucose-restricted CD8⁺ T cell function by promoting histone acetylation and chromatin accessibility (Qiu Jing et al., *Cell Reports*, 2019), improving memory CD8⁺ T cell function by expanding their acetyl-CoA pool (Maria L et al., *Immunity*, 2016), and modulating cellular metabolism and mTOR activity (Luu Maik et al., *Sci. Rep.*, 2018). According to these comments, we have now performed additional experiments for better understanding the mechanisms of how acetate regulate tumor immunity. We have also now added appropriate discussions to incorporate these references.

(1) To explore how acetate impacts CD8⁺ T cell function in co-culture conditions, we performed RNA sequencing on CD8⁺ T cells co-cultured with 4T1 cells treated with or without sodium acetate. Of the 29,956 gene transcripts expressed, only 1828 genes (6.1%) differed significantly between the groups, and no significant cluster was observed between groups (Fig. 6j). Of the 28 effector-, memory-, and exhaustion-related genes of CD8⁺ T cells^[1], the mRNA levels of *Gzma*, *Lta*, *Tnf*, *Batf3*, *Eomes*, *Foxo1*, *Id2*, *Irf4*, *Tbx21*, *Ii12rb2*, *Cd69*, *Elf4*, *Nfatc3*, *Tcf3*, *Il7r*, and *Il27ra* showed significant difference, but the log₂ fold-change in gene expression > 1 was only observed for *Gzma* and *Lta* (Fig. S5j). Neither glycolysis nor TCA cycle genes clustered differently between acetate-treated and control CD8⁺ T cells. Furthermore, except for the decreased expression of some genes in the glycolytic pathway (*Gpi1*, *Hk1*, *Pgam1*, *Pgk1*, *Pkm*, *Tpi1*, and *Pfkfb*) (Fig. S5k), acetate

treatment had no significant influence in TCA cycle genes at the individual gene level (Fig. S51). Several monocarboxylate transporters including SLC16A1 and SLC16A3 were involved in acetate transport^[2-3]. At the mRNA level, we observed that the expression of *Slc16a1* but not *Slc16a3*, was increased in acetate-exposed CD8⁺ T cells (Fig. 6k-6l). Previous studies have demonstrated that acetate may enhance IFN- γ production in a manner dependent on acetyl-CoA synthetase (ACSS) and ATP citrate lyase (ACLY)^[2-3]. In our study, we observed the upregulation of mRNA levels of *Acss1*, *Acss2*, and *Acly* in acetate-exposed CD8⁺ T cells (Fig. 6l). Isotopic tracing studies revealed that acetate contributed to the production of citrate, oxoglutarate, fumarate, and malate (Fig. 6m). Moreover, exogenous acetate increased the pool of cellular acetyl-CoA available to CD8⁺ T cells (Fig. 6n). Together, these data suggest that, in conditions of CD8⁺ T cells co-culture with cancer cells, acetate treatment may boost the activity of CD8⁺ T cells via restoring the acetyl-CoA pool and subsequently augment IFN- γ production.

Fig. 6. (j) Heatmap (left panel) and MA-plot (right panel) of control and acetate-exposed CD8⁺ T cells under co-culture with 4T1 cells transcriptome (n=5 per group). (k) Pathway of acetate uptake and metabolism in CD8⁺ T cells. (l) mRNA expression of the solute carrier receptors *Slc16a1*, *Slc16a3*, *Acss1*, *Acss2*, and *Acly* in control and acetate-exposed CD8⁺ T cells under co-culture with 4T1 cells as determined by qPCR (n=4 per group). (m) Metabolic tracing analysis of CD8⁺ T cells under co-culture with 4T1 cells after exposure to ¹³C-acetate. The x-axis shows the number of ¹³C per respective metabolite. Depicted are pooled data from two independent experiments with cells from n=3 mice each. (n) Acetyl-CoA concentration in CD8⁺ T cell lysates after co-culture with 4T1 cells with or without acetate (n=5 per group).

Fig. S5. Heatmaps for gene expression of transcription factors regulating CD8⁺ T cell effector, memory and exhaustion (j), glycolysis (k) and TCA cycle (l) of control and acetate-exposed CD8⁺ T cells under co-culture with 4T1 cells (n=5 per group).

References:

[1] He, Y., et al. Gut microbial metabolites facilitate anticancer therapy efficacy by modulating cytotoxic CD8(+) T cell immunity. *Cell Metab* 33, 988-1000 e1007 (2021).
 [2] Qiu, J., et al. Acetate Promotes T Cell Effector Function during Glucose Restriction. *Cell Rep* 27, 2063-2074 e2065 (2019).
 [3] Balmer., et al. Memory CD8 + T cells require increased concentrations of acetate induced by stress for optimal function. *Immunity* 44, 1312-1324.

(2) As you noted, various short-chain fatty acids exhibit distinct effects. For example, research has shown that specific microbial systemic short-chain fatty acids, such as butyrate and propionate, but not acetate, can impact the anti-CTLA-4 anti-tumor effect in mouse models and patients with metastatic melanoma receiving ipilimumab treatment (Systemic short chain fatty acids limit antitumor effect of CTLA-4 blockade in hosts with cancer, *Nature Comms.*,2020).

We conducted experiments to investigate the effects of propionate and butyrate on the survival and cytotoxic capacity of CD8⁺ T cells, with and without co-culture with 4T1 cells. Consistent with previous results, our findings showed that, unlike sodium propionate and sodium butyrate, sodium acetate had no direct impact on the survival and functionality of CD8⁺ T cells (Fig. S5e and Fig. S5f). By contrast, sodium acetate treatment, along with sodium propionate and sodium butyrate, increased the survival and production of IFN- γ in CD8⁺ T cells co-cultured with 4T1 cells (Fig. 6h-6i and Fig. S5h-S5i).

Fig.S5. (e) Percentage of CD8⁺ T cell death after sodium propionate and sodium butyrate treatment and IFN- γ abundance of CD8⁺ T cells (n=5 per group). Percentage of CD8⁺ T cell death in co-culture with 4T1 cells after sodium propionate and sodium butyrate treatment and IFN- γ abundance of CD8⁺ T cells (n=5 per group).

(3) According to your suggestions, we have also performed metabolic tracing studies on CD8⁺ T cells co-cultured with or without cancer cells under high or low glucose conditions. Our findings revealed that, in low glucose conditions, acetate contributes to the increased production of oxoglutarate, but not fumarate, malate, or citrate in CD8⁺ T cells without co-culture (Fig. a). In contrast, we observed a substantial contribution of acetate to the elevated production of oxoglutarate, fumarate, and malate in low glucose conditions when CD8⁺ T cells were co-cultured with cancer cells (Fig. b). These results suggest that acetate supplementation might partially compensate for the competitive loss of these crucial glycolytic intermediates in CD8⁺ T cells within the tumor microenvironment in which T cells are likely in short of glucose usage. However, the effects of acetate in restoring the functions and abundancies of CD8⁺ T cells co-cultured with cancer cells are unlikely dependent of glucose availability (Fig. c). On the basis of these results, we prefer not to include these data to the revised manuscript but provided here to address the reviewer's concerns.

Figures. (a) Metabolic tracing analysis on CD8⁺ T cells followed by exposure to ¹³C-acetate (5 mM) under varying glucose concentrations (11 and 1 mM, respectively). The x-axis represents the ¹³C abundance per metabolite. The presented data represents pooled results from two separate experiments, each performed with cells obtained from n=3 mice. (b) Metabolic tracing analysis on CD8⁺ T cells co-cultured with 4T1 cells, followed by exposure to ¹³C-acetate (5 mM) under varying glucose concentrations (11 and 1 mM, respectively). (c) The percentage of CD8⁺ T cell death and

the abundance of IFN- γ after co-culturing with 4T1 cells in the presence or absence of acetate (5 mM), under different glucose concentrations (n=5 per group).

2. The methods used in this study are sound. The authors have included both, in vivo experiments and translational data. However, there is too much focus on a possible link between depression in breast cancer patients and *Blautia*-acetate axis. In the experimental model, the chronic stress enhances breast cancer progression. In contrast, in humans, diagnosis of cancer can lead to serious depression. In terms of mechanistic understanding, the data from the Fig. 7 are overinterpreted. For the Fig. 7b (IHC), authors analyzed a cohort of only 5 breast cancer patients without depression and 8 depressed cancer patients. I am not sure if the Fig. 7 really supports the conclusions and claims. For their primary statements, authors should use a larger group of patients and should also more focus on the mechanistic part of the study in the murine experimental model of the stress-promoted cancer progression. It would be important to generate a mutant strain of *Blautia* *coccoides* that is not able to produce acetate to analyze *Blautia*-acetate-mediated effects in vivo. It would be also important to establish a mechanistic link between *Blautia*-acetate deficiency and stress hormones that are known to promote tumorigenesis in order to strengthen the current claims.

Response: Thanks for all these comments.

(1) We agree with the reviewer's comment and have therefore increased the sample size for the breast cancer human cohort. We have also obtained the prognosis information of both depressed and non-depressed breast cancer patients. Our findings suggest that breast cancer patients with depression display lower levels of acetate, reduced populations of tumor-infiltrating CD8⁺ T cells, and an elevated risk of metastasis. These results are visually presented in Figure 7 of the revised manuscript. We have also incorporated these updated results into the abstract and discussion sections of the revised manuscript.

Fig. 7. Acetate levels correlate with CD8⁺ T cells in breast cancer patients. (a) Serum acetate levels in breast cancer patients with and without comorbid depression (n=28 and n=24, respectively). (b) Immunohistochemical staining of CD8⁺ T cells on breast cancer tissue samples obtained from patients with and without comorbid depression (n=17 and n=11, respectively). (c) Spearman correlation analysis between serum acetate levels and the count of tumor-infiltrating CD8⁺ T cells (n=28). (d) The relationship among serum acetate levels, count of tumor-infiltrating CD8⁺ T cells, and the depression score of breast cancer patients. Each sample (n=28) was represented by a dot, with the dot's size indicating the CD8⁺ T cell count and its color representing the serum acetate levels. (e) Kaplan-Meier metastasis-free survival curves for breast cancer patients with and without comorbid depression (n=28 and n=24, respectively). Statistical significance was determined by unpaired two-tailed Student's t-test. Significance levels are denoted as * $p < 0.05$; ** $p < 0.01$; *** $p < 0.001$; **** $p < 0.0001$.

(2) *Blautia* is a well-known acetogen, and our study demonstrated that both *Blautia* and acetate supplementation can reverse stress-promoted breast cancer progression. As you mentioned, generating a mutant strain of *Blautia* that cannot produce acetate would be crucial for analyzing the in vivo effects of *Blautia*-acetate. However, one of the main challenges in modifying *Blautia*'s

metabolism is the lack of effective genetic modification tools due to its thick cytoderm, which makes it difficult for current gene-editing techniques to penetrate and mutate target genes involved in acetate production. Additionally, we were unsuccessful in finding an antibiotic to selectively deplete *Blautia*.

(3) To address this concern, we used shotgun metagenomics to determine the alteration of the enzymes of *Blautia* involved in acetate metabolism. Our metagenomic data indicated that the Wood-Ljungdahl pathway, which is the main pathway generating acetate by *Blautia*, was significantly downregulated, suggesting decreased acetyl-CoA capacity (Fig. 5i-5j). We observed that nearly all the key genes involved in CO₂ fixation (K00198, K15022), formyl-tetrahydrofolate ligase (K01491, K00297), and acetyl-CoA synthase (K01438) were significantly reduced (Fig. 5k), supporting the reduction of acetyl-CoA biosynthesis upon chronic stress. Further analysis revealed striking changes in pyruvate metabolism including phosphoacetylase (K00625) and acetate kinase (K00925) in stressed tumor mice (Fig. 5k). To identify the bacterial genera involved in Wood-Ljungdahl pathway, we conducted an analysis of species contributing to K00198 from the top 50 bacterial genera (Fig. S4f), and it was determined that the crucial K00198 was present in *Blautia*, *Lachnoclostridium*, *Ruminococcus*, *Clostridium*, and *Oscillibacter* genera. Taken together, these findings support that, under conditions of chronic stress, the decreased abundance of *Blautia* is more likely in contribution to the reduction of acetate production, and that the disruption of *Blautia*-acetate axis is likely involved in the pathological progression of breast cancer promoted by chronic stress.

Fig. 5. (i) Linear discriminative analysis (LDA) Effect Size (LEfSe) analysis between tumor group and stressed tumor group in KEGG modules. (j) Overview of the Wood-Ljungdahl pathway: K00198: anaerobic carbon-monoxide dehydrogenase catalytic subunit; K15022: formate dehydrogenase (NADP⁺) beta subunit; K01491: methylenetetrahydrofolate dehydrogenase (NADP⁺) / methylenetetrahydrofolate cyclohydrolase; K00297: methylenetetrahydrofolate reductase (NADH); K14138: acetyl-CoA synthase; K00625: phosphate acetyltransferase; K00925: acetate kinase. (k) The relative abundance KOs associated with Wood-Ljungdahl pathway (n=5 per group).

Fig. S4. (f) Barplot of species and functional contribution analysis, highlighting the top 50 genus contributing the k00198.

(4) We agree with this comment. Stress hormones are known to play a role in depression facilitated tumor development, and the gut microbiota influences the production of stress hormones through the gut-brain axis and microbial metabolites^[1-2]. And thus it is likely that the stress hormones may connect to microbiota-metabolites signals in pathological development of depression facilitated cancer progression. Although it is an intriguing question whether *Blautia*-acetate deficiency is causally linked to the secretion of stress hormones, this topic is beyond the scope of the present study. Nonetheless, we appreciate the suggestion and have added some discussions about the potential links between gut microbiota and stress hormones.

References:

[1] John F Cryan et al. The Microbiota-Gut-Brain Axis. *Physiol Rev.* 2019 Oct 1;99(4):1877-2013.
 [2] Kieran Rea, Timothy G Dinan, John F Cryan. The microbiome: A key regulator of stress and neuroinflammation. *Neurobiol Stress.* 2016 Mar 4;4:23-33.

3. Of note, acetate is most abundant SCFA, which is produced by most of the enteric bacteria. Why should only acetate generated by *Blautia*, and not that produced by other bacteria, play an important role?

Response: Thanks for this instructive comment. We agree with this comment that there are other potential acetate-producing bacteria in addition to *Blautia* might be also applied to compensate for

Blautia reduction. To address this concern, we have added some discussions: Although there are many enteric bacteria that produce acetate^[1-2], *Blautia* is one of the most abundant genera in the gut, regardless of race^[3-5], and is a key member of the Lachnospiraceae family that accounts for 50% of the total intestinal microbiota^[6]. Moreover, a previous study examining the acetate-producing capabilities of 110 Lachnospiraceae strains found that 91 strains produced acetate, and the top five acetate producers were all from *Blautia*^[7-8]. Our study revealed that *Blautia* was significantly reduced in conditions of depression, indicating that it is the most important acetate producer in the gut. In light of this evidence, we propose that the supplementation with *Blautia* is an effective approach to compensate for the reduction of acetate production in the gut.

References:

- [1] Koh A, De Vadder F, Kovatcheva-Datchary P, Bäckhed F. From Dietary Fiber to Host Physiology: Short-Chain Fatty Acids as Key Bacterial Metabolites. *Cell*. 2016 Jun 2;165(6):1332-1345. doi: 10.1016/j.cell.2016.05.041. PMID: 27259147.
- [2] Petra Louis, Georgina L Hold, Harry J Flint. The gut microbiota, bacterial metabolites and colorectal cancer. *Nat Rev Microbiol*. 2014 Oct;12(10):661-72.
- [3] Nishijima, S. et al. The gut microbiome of healthy Japanese and its microbial and functional uniqueness. *DNA Res*. 23, 125–133 (2016).
- [4] Bamberger, C. et al. A walnut-enriched diet affects gut microbiome in healthy Caucasian subjects: a randomized, controlled trial. *Nutrients* 10, 244 (2018).
- [5] Zhang, W. et al. Gut microbiota community characteristics and disease-related microorganism pattern in a population of healthy Chinese people. *Sci. Rep.* 9, 1594 (2019).
- [6] Biddle A, Stewart L, Blanchard J, Leschine S. Untangling the genetic basis of fibrolytic specialization by lachnospiraceae and ruminococcaceae in diverse gut communities. *Diversity*. 2013;5(3):627–640. doi:10.3390/d5030627.
- [7] Abdugheni, R., Wang, W.-Z., Wang, Y.-J., Du, M.-X., Liu, F.-L., Zhou, N., Jiang, C.-Y., Wang, C.-Y., Wu, L., Ma, J., Liu, C., Liu, S.-J. Metabolite profiling of human-originated Lachnospiraceae at the strain level. *iMeta*. 2022 Dec; 1(4): e58.
- [8] Vacca M, Celano G, Calabrese FM, Portincasa P, Gobbetti M, De Angelis M. The Controversial Role of Human Gut Lachnospiraceae. *Microorganisms*. 2020 Apr 15;8(4):573. doi: 10.3390/microorganisms8040573.

5. The Fig. 2 may suggest that reduced acetate levels would have opposite effect as compared to other figures. It is known that vancomycin exposure leads to depletion of acetate-producing bacteria in the gut and that mice treated with antibiotics produce less acetate. This figure suggests that gut microbiota drives the stress-associated breast cancer progression. How can the authors explain this observation?

Response: Breast cancer growth was found to be promoted in conditions of chronic stress, but this effect was absent in mice treated with ABX (Fig. 2), indicating that gut microbiota plays a role in chronic stress-induced breast cancer progression. The ABX treatment protocol is commonly used to preliminarily determine whether a phenotype is correlated with gut microbiota. It should be noted that results obtained from this protocol should not be compared with those from other experimental protocols. In this case, both the tumor and stressed tumor groups were treated with ABX, which may have led to comparable levels of acetate in both groups and hence comparable tumor growth.

However, we acknowledge that other bacteria that directly promote cancer progression and are eliminated by ABX treatment may also be involved in this complex setting.

6. Collectively, the authors could try to explore more mechanistic findings in mice, instead of strong focus on patients in the abstract and discussion. Does the data generated with samples from depressed breast cancer patients support all conclusions and claims? It would be of great benefit if the authors can test their hypothesis in another and larger cohort of depressed cancer patients (across various cancer types).

Response: We have further explored mechanistic findings in mice through the use of shotgun metagenomics, metabolic tracing analysis, and RNA sequencing analysis in the revised manuscript. Our study indicates that breast cancer patients with depression exhibit lower levels of serum acetate and a lower abundance of tumor-infiltrating CD8⁺ T cells, along with an increased risk of metastasis. These findings suggest that the reduction of acetate and CD8⁺ T cells may underlie tumor progression in depressed breast cancer patients. The human patient findings are consistent with those obtained from animal studies, suggesting that the compromised *Blautia*-acetate axis is likely involved in depression-promoted breast cancer progression in humans. However, further comprehensive studies are needed to confirm the causal link between *Blautia* and acetate in depression-associated breast cancer in clinical settings. Additionally, this study only investigated the role and mechanism of depression in promoting breast cancer. In the future, we will follow your suggestions and further explore whether the *Blautia*-acetate axis plays a role in other depression-cancer models and patients.

Minor points:

1. Line 787 (Figure legend of the Fig 7.) the word “corrected” should be replaced by the word “correlated”.

Response: We have corrected it in the revised manuscript.

2. In the Figs. 3c and 3f, the authors use the terminology: “*Blautia* / relative expression”. I do not think that bacteria can be “expressed”? I would suggest the words: “relative abundance”.

Response: Based on your professional suggestions, we have revised it in the manuscript.

REVIEWERS' COMMENTS

Reviewer #1 (Remarks to the Author):

The authors have done a good job of adding additional experiments and making other modifications in their paper, all of which have improved their manuscript.

Reviewer #2 (Remarks to the Author):

My questions have all been properly addressed. I am satisfied with the answers.

Reviewer #4 (Remarks to the Author):

The authors have satisfactorily addressed most of my concerns.